# Pellino1 regulates reversible ATM activation via NBS1 ubiquitination at DNA double-strand breaks

Geun-Hyoung Ha[1], Jae-Hoon Ji[2], Sunyoung Chae[3], Jihyun Park[4], Suhyeon Kim[1], Jin-Kwan Lee[4], Yonghyeon Kim[5], Sunwoo Min[2,5], Jeong-Min Park[6], Tae-Hong Kang[6], Ho Lee[7], Hyeseong Cho[2,5] & Chang-Woo Lee[1,4]

DNA double-strand break (DSB) signaling and repair are critical for genome integrity. They rely on highly coordinated processes including posttranslational modifications of proteins. Here we show that Pellino1 (Peli1) is a DSB-responsive ubiquitin ligase required for the accumulation of DNA damage response proteins and efficient homologous recombination (HR) repair. Peli1 is activated by ATM-mediated phosphorylation. It is recruited to DSB sites in ATM- and γH2AX-dependent manners. Interaction of Peli1 with phosphorylated histone H2AX enables it to bind to and mediate the formation of K63-linked ubiquitination of NBS1, which subsequently results in feedback activation of ATM and promotes HR repair. Collectively, these results provide a DSB-responsive factor underlying the connection between ATM kinase and DSB-induced ubiquitination.

[1] Department of Molecular Cell Biology, Samsung Medical Center, Sungkyunkwan University School of Medicine, Suwon 16419, Republic of Korea. [2] Genomic Instability Research Center, Ajou University School of Medicine, Suwon 16499, Republic of Korea. [3] Institute of Medical Science, Ajou University School of Medicine, Suwon 16499, Republic of Korea. [4] Department of Health Sciences and Technology, SAIHST, Sungkyunkwan University, Seoul 06351, Republic of Korea. [5] Department of Biochemistry and Molecular Biology, Ajou University School of Medicine, Suwon 16499, Republic of Korea. [6] Department of Biological Science, Dong-A University, Pusan 49201, Republic of Korea. [7] Graduate School of Cancer Science and Policy, Research Institute, National Cancer Center, Goyang 10408, Republic of Korea. These authors contributed equally: Geun-Hyoung Ha and Jae-Hoon Ji. Correspondence and requests for materials should be addressed to H.C. (email: hscho@ajou.ac.kr) or to C.-W.L. (email: cwlee1234@skku.edu) or to J.H.J. (email: jij@ajou.ac.kr)

f DNA double-strand breaks (DSBs) are impaired, they cause loss of genetic information by mutations or gross chromosomal rearrangements, both of which are hallmarks of cancer cells[1]. DSBs trigger DNA damage response (DDR), which regulates specialized cellular processes such as cell cycle checkpoint and promotes activation of DNA repair pathways. Mammalian cells employ two major DNA repair pathways, homologous recombination (HR) and non-homologous end joining (NHEJ), thereby suppressing genomic instability[2–4]. HR repair can be error-free, which requires a homologous template such as a sister chromatid, whereas NHEJ joins the two ends of a DSB through a process largely independent of homology[1–4]. DSB is detected by sensor proteins that can trigger activation of proximal kinases such as ATM and ATR[3,4]. These kinases in turn activate a series of more distal kinases such as Chk1 and Chk2, which can phosphorylate and regulate a number of protein effectors of the checkpoint and DDRs[5]. Ku70/Ku80 heterodimer is also a specialized DSB sensor recruited to DSBs[6]. Ku complex results in recruitment of DNA-PKcs, which is activated by the presence of free DNA ends to initiate NHEJ repair process DSBs[6].

Ataxia telangiectasia is caused by defects in Ataxia telangiectasia mutated (*ATM*) gene. It is related to Nijmegen breakage syndrome (NBS)[7,8]. NBS is an autosomal recessive genetic disorder characterized by immunodeficiency, microcephaly, growth retardation, and a high frequency of lymphoid malignancies[8,9]. Cells from NBS patients exhibit highly elevated sensitivity to ionizing radiation (IR), chromosome instability, and abnormal cell cycle checkpoints[10]. NBS1 protein encoded by *NBS1* gene can interact with several functional proteins including ATM. These interactions are vital for various DDRs. Interaction between NBS1 and phosphorylated histone H2AX is responsible for recruitment of NBS1 to DSB sites[11]. Germline mutations in the *NBS1* gene can lead to cancer-prone developmental disorder NBS[12–14]. Mediator of DNA-damage checkpoint 1 (MDC1) is another binding partner of NBS1. When MDC1 is phosphorylated by casein kinase 2, it can interact with NBS1. This interaction may be important for the accumulation of NBS1 at DSB sites[15,16]. DSB repair protein MRE11, a human ortholog of yeast meiotic recombination 11[17], also directly interacts with RAD50, another DNA repair protein[18]. These proteins (MRE11/RAD50/NBS1, MRN) form a stable complex that allows nuclear localization of molecules and facilitates their functions in DDR pathways and HR repair. As a part of the MRN complex, NBS1 exhibits a pleiotropic role in DNA repair.

Ubiquitination of cellular proteins is versatile and reversible. It is integrated into the dynamic and complex cellular process of DSB repair[1]. Lysine (K) 48- and K11-linked ubiquitin chains are major signals for protein degradation via the 26S proteasome, whereas non-proteolytic ubiquitination has an important regulatory role in DSB signaling and repair[1]. In particular, K63-linked chains are instrumental in recruiting proteins to DSB sites. RNF8 and RNF168 are ubiquitin ligases extensively studied in the DDR pathway. In DDR, phosphorylated H2AX recruits MDC1 and its partner RNF8[19,20]. RNF8 ubiquitinates histones that can initiate subsequent recruitment of RNF168. RNF168 further ubiquitinates histones around the damage site[21,22]. This serves as a platform for downstream DNA repair proteins such as BRCA1 and 53BP1[23,24]. Therefore, integrated mechanism by ubiquitination regulates accurate and efficient processes of DSB repair.

Pellino (Peli) proteins are known as signal-responsive ubiquitinligases. They have emerged as important factors in innate immunity, tumorigenesis, and potentially metabolism[25,26]. Recent studies have unveiled a critical role of Peli1 in activating receptor signaling such as Toll-like receptor and/or T-cell receptor (TCR) signaling to mediate transcriptional regulation of proinflammatory genes[27]. Indeed, loss of Peli1 can lead to hyperactivation and nuclear accumulation of c-Rel in response to TCR-CD28 signaling, contributing to the development of autoimmune disease[28,29]. Notably, Peli proteins include forkhead-associated (FHA) domains, which are small protein modules that can recognize phosphothreonine epitopes on proteins[30]. It becomes clear that FHA domain-mediated phospho-dependent assembly of protein complexes has a wide range of regulatory mechanisms. Interestingly, FHA domains are also present in DNA-damage checkpoint kinase Chk2, Dun1, and NBS1. FHA domains of these proteins play a critical role in integrating upstream signals[31]. Taken together, these findings suggest that Peli proteins have a scaffolding function to facilitate complex formation of DNA-damage-responsive proteins. In this study, we show that Peli1 is likely to be an immediate DSB-responsive ubiquitin ligase that is activated by ATM-mediated phosphorylation, subsequently promoting the accumulation of ATM and MRN complex at DSB sites via NBS1 ubiquitination.

## Results

**Peli1 is recruited to DNA DSB sites.** Initially, we found that loss of Peli1 led to extensive rates of DSBs, presumably both chromatid and chromosome breaks, in mitotic spreads (Fig. 1a, b). Thus, we investigated whether Peli1 may be involved in DDR. We determined whether Peli1 responds to a DSB signal. Our results revealed that IR clearly induces Peli1 in primary murine embryonic fibroblast (MEF) cells (Fig. 1c). Importantly, Peli1 was translocated to DSB sites and colocalized with γH2AX after IR. However, in undamaged MEF cells, Peli1 was distributed throughout the cytoplasm and some were found in the nucleus (Fig. 1d). We also found that Peli1 is recruited to and accumulated at laser stripes as readily as 53BP1 (Fig. 1e). Interestingly, other Peli family proteins such as Peli2, Peli3α, and Peli3β were also recruited to DSB sites (Supplementary Fig. 1), indicating that such phenomenon might be common for Peli family proteins. DSBs not only can activate ATM pathway but also can turn on ATR pathway. During nuclease-mediated resection of DSBs, replication protein A (RPA)-coated single-stranded DNA (ssDNA) as a structural platform for ATR activation can be generated[32]. To determine whether Peli1 can also activate the ATR pathway, we treated cells with UV or hydroxyurea to activate ATR pathway by generating RPA-coated ssDNA. Unlike regulation for ATM activation, Peli1 is not required for ATR activation (Supplementary Fig. 2). In addition, depletion of Peli1 showed no impairment in nucleotide excision repair activity against UV lesions (Supplementary Fig. 2).

To extend our observation, we determined whether Peli1 is recruited to a single DSB site. We took advantage of a DSB reporter system that used LacI-FokI nuclease fusion protein to create a DSB within a single genomic locus in U2OS cells (U2OS-DSB reporter)[33]. To detect a single focus at a DSB in single cell, YFP-MRE11 or GFP-Peli1 was transfected with mCherry-LacI-FokI. Interestingly, similar to YFP-MRE11, GFP-Peli1 was colocalized with FokI nuclease-induced DSB site (Fig. 1f). We also confirmed colocalization of endogenous γH2AX, 53BP1, RPA32, RAD51, and Peli1 with FokI. Their colocalizations were identified from G1 to S/G2 phase of cell cycle (Fig. 1g and Supplementary Fig. 3a, b). To investigate the function of Peli1 at sites of DNA damage, we applied a kinetic assay using micro-irradiation-generated DSBs coupled with live imaging of protein redistribution[34]. Through this approach, we found that GFP-Peli1 is rapidly and robustly accumulated in micro-irradiated regions. Signs of local GFP-Peli1 were detectable in <60 s, similar to that of GFP-NBS1 in DSB sites (Fig. 1h, i). GFP-Peli1 accumulation occurred faster than GFP-MDC1 accumulation (Fig. 1h, i).

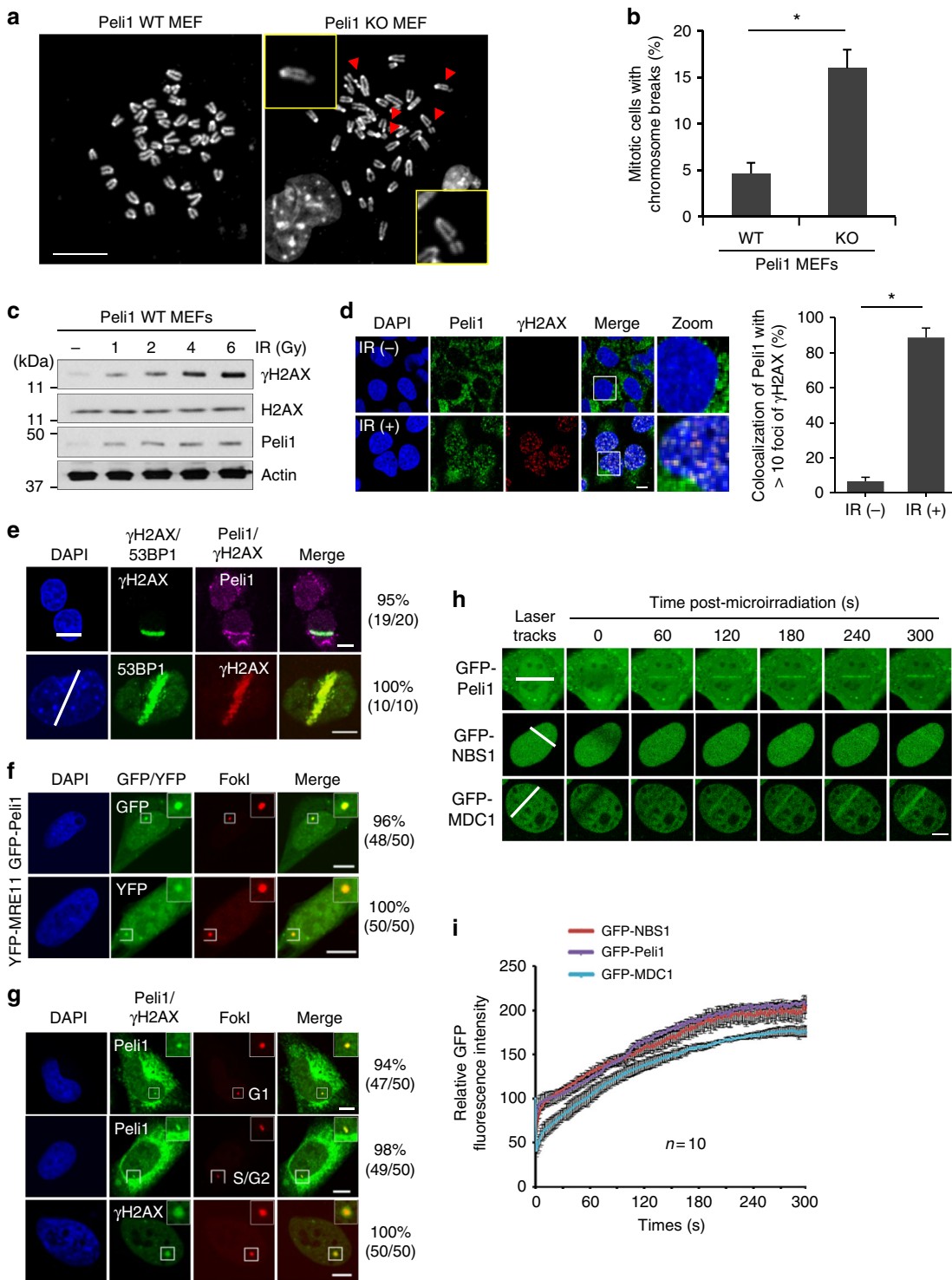

Collectively, these results suggest that Peli1 is an early response protein at DSB sites.

**FHA domain-mediated recruitment of Peli1 at DSB sites**. Next, we identified regions within Peli1 important for its accumulation at DSB sites. We used a series of green fluorescent protein (GFP)-tagged Peli1 mutants (Fig. 2a, b) and monitored the recruitment to DSB sites. Peli1 mutants with FHA1 and/or FHA2 domain deleted (ΔFHA1, ΔFHA2, and C) lost their ability to be recruited to laser tracks, whereas wild-type (WT) and ring-like domain-deleted mutant (ΔC) retained their ability to bind to DSB sites

(Fig. 2c). Furthermore, we transfected GFP-Peli1 WT and mutants into U2OS reporter cells. Immunostaining analysis revealed that Peli1 WT and ΔC were markedly enriched in FokI-induced DSB site. However, Peli1 ΔFHA1, ΔFHA2, and C mutants almost completely lost their ability to be recruited to the DSB site (Fig. 2d). Signals of GFP-Peli1 ΔFHA1 and GFP-Peli1 ΔFHA2 were mainly detected in the cytoplasm rather than the nucleus, as low intensity of GFP-Peli1 ΔFHA1 and GFP-Peli1 ΔFHA2 in the nucleus may affect their translocation at laser stripes. Thus, we further examined recruitments of GFP-Peli1 ΔFHA1 and GFP-Peli1 ΔFHA2 with additional heterologous

**Fig. 1** Peli1 is recruited to DNA double-strand break sites. **a** Mitotic spreads of primary Peli1 wild-type (WT) and Peli1 knockout (KO). MEF cells were arrested in metaphase with 0.1 mg/ml of colchicine, fixed, and visualized by DAPI staining. Red arrow designates chromatid and chromosome breaks. Scale bars, 5 μm. **b** Quantification of the number of chromatid and chromosome breaks per cell evaluated in 50 cell metaphase spreads. Student's *t*-test was used for statistical analyses. **c** Peli1 WT MEF cells were treated with IR at different doses (0, 1, 2, 4, and 6 Gy). At 1 hr after IR, cells were collected and then subjected to immunoblotting with Peli1, γH2AX, H2AX, and actin antibodies. **d** Peli1 WT MEFs were either control treated or treated with IR (10 Gy) and allowed to recover for 1 h before processing for Peli1 and γH2AX immunostaining (left). Scale bars, 10 μm. Quantitative analysis for colocalization of Peli1 with >10 foci of γH2AX without IR or with IR. Plotted values represent mean ± SEM of more than 300 individual cells. Student's *t*-test was used for statistical analyses. **e** Peli1 WT MEF cells were subjected to laser micro-irradiation. After 10 min, cells were co-stained with γH2AX or 53BP1 and Peli1 antibodies. The white lines with DAPI staining indicated laser stripes. Scale bars, 10 μm. **f** mCherry-LacI-FokI expressing plasmid was cotransfected with GFP-Peli1 (upper panels) or YFP-MRE11 (positive control, lower panels) into U2OS-DSB reporter cells. Scale bars, 10 μm. **g** mCherry-LacI-FokI plasmid was transfected into U2OS-DSB reporter cells. At 48 h post transfection, cells were immunostained with Peli1 or γH2AX antibodies. The number of FokI focus determines the cell cycle phase. Single FokI (upper panel) represents G1 phase while double FokI represents S/G2 phase of cell cycle. Scale bars, 10 μm. **h**, **i** U2OS cells were transfected with GFP-Peli1, GFP-NBS1, or GFP-MDC1 expressing plasmid. At 48 h, cells were subjected to laser micro-irradiation. Laser stripes were examined at indicated time point (**h**). The intensity of each laser stripe was determined by averaging values from ten cells at each time point and graphed (**i**). Scale bars, 10 μm

nuclear localization signal (NLS) (GFP-Peli1-NLS-ΔFHA1 and GFP-Peli1-NLS-ΔFHA2, respectively) to laser stripes or mCherry-LacI-FokI site. The results showed that these mutants are unable to translocate to laser stripes or FokI site (Fig. 2c, d and Supplementary Fig. 4). Therefore, we propose that Peli1 FHA domains, but not its E3 ligase domain, are essential for targeting Peli1 to DSB sites.

We next determined whether the involvement of Peli1 in DDR is dependent on its FHA domain in vivo. To test this possibility, we generated Peli1 mutant mice by a conventional targeting strategy in which coding exon 4 was replaced with a puromycin cassette, resulting in Peli1 exon 4 deletion (ΔE) mice with FHA2 domain specifically deleted (hereafter Peli1 ΔE4; Fig. 2e and Supplementary Fig. 5a, b). These generated homozygous Peli1^ΔE4/ΔE4 (Peli1 ΔE4) mice did not show apparent abnormalities in growth or survival compared with Peli1 WT (Peli1^+/+) mice at 6 months of age. However, if Peli1 ΔE4 mice were bred for more than 8 months, the survival rate dropped gradually as shown in Peli1 homozygotes (Supplementary Fig. 5c, d). To investigate the effect of Peli1 FHA2 deficiency on response to DNA damage, we isolated primary MEFs and found that Peli1 ΔE4 MEFs displayed marked reduction in γH2AX level (Fig. 2f) with reduced and diffused recruitments of γH2AX and phospho-ATM to DSB foci following IR (Fig. 2g), indicating that Peli1 FHA domain can mediate the recruitment of Peli1 to DSB sites.

**ATM-dependent accumulation of Peli1 at DNA damage sites.** We next determined the mechanism by which Peli1 is recruited to DNA damage sites. We considered whether Peli1 recruitment is dependent on upstream kinases (such as ATM, ATR, or DNA-PKs), poly(ADP-ribose) polymerase (PARP), and/or sumoylation pathways. Cells were treated with selective inhibitors targeting ATM (ATMi, KU55933), ATR (ATRi, VE-831), DNA-PKs (DNA-PKi, KU57788), PARP (PARPi, PJ34), or small ubiquitin-like modifier (SUMO) (SUMOi, 2D-08), and then the ability of GFP-Peli1 to be recruited to laser tracks after micro-irradiation was examined (Fig. 3a, b). Interestingly, we found that treatment with ATM inhibitor sharply reduces GFP-Peli1 recruitment to DSB sites. However, treatment with other inhibitors has no effect on its recruitment (Fig. 3a, b). Similarly, the recruitment of endogenous Peli1 to DSB sites is sharply reduced in ATM-deficient MEFs compared with that in ATM WT MEFs (Fig. 3c). These results indicate that the recruitment of Peli1 to DSB sites appears to be dependent on ATM.

Notably, there are three putative ATM/ATR substrate consensus phosphorylation sites (SQ/TQ) in Peli1 (Ser 121, Thr 127, and Ser 380). To understand whether these SQ/TQ motifs of

Peli1 might be responsible for ATM-dependent accumulation at DSB sites, we mutated Ser 121 (S121) and Thr 127 (T127) to alanine in the context of full-length GFP-fused Peli1 (Fig. 3d). Importantly, phosphorylation of Peli1 WT responding to IR was clearly observed with phospho-specific SQ/TQ antibody that could specifically recognize proteins phosphorylated on SQ/TQ motifs. However, mutations of Peli1 at S121 and T127 residues resulted in complete loss of phosphorylation as indicated by SQ/TQ antibody (Fig. 3e). Interestingly, IR-induced phosphorylation of Peli1 on SQ/TQ motifs seemed to be correlated with IR-induced Peli1 ubiquitination activity (Fig. 3f). Next, we examined whether SQ/TQ motifs are important for the recruitment of Peli1 to DSBs. GFP-Peli1 S121A/T127A mutant showed a clear reduction in its recruitment to DSBs compared with GFP-Peli1 WT following micro-irradiation (Fig. 3g–i). These results indicate that ATM-mediated phosphorylation of Peli1 on SQ/TQ motifs is important for its ubiquitination activity and subsequent recruitment to DSB sites.

**γH2AX-mediated recruitment of Peli1 to DSB sites.** To delineate whether Peli1 might participate in the established DNA-damage signaling cascade, we examined the distribution of GFP-Peli1 in cells depleted of various DNA-damage signaling factors such as H2AX, MDC1, MRE11, and CtIP following micro-irradiation. Depletion of H2AX resulted in lost ability to recruit GFP-Peli1 to laser tracks, whereas depletion of MRE11, CtIP, or MDC1 retained this capability (Fig. 4a–c). Similarly, micro-irradiation-induced accumulation of endogenous Peli1 was significantly reduced in H2AX-deficient HeLa cells compared with that in WT HeLa cells (Fig. 4d). The recruitment of GFP-Peli1 was also sharply reduced in H2AX knockout (KO) cells, but not in H2AX WT cells (Fig. 4e, f), indicating that H2AX is required for the recruitment of Peli1 at DSB sites.

γH2AX, the phosphorylated form of H2AX, is involved in the initial steps of chromatin decondensation after DSB. Thus, γH2AX is required for DNA-damage signal amplification and subsequent accumulation of various DDR proteins at DSB sites[3,35]. In this regard, we generated expression plasmids encoding FLAG-tagged H2AX WT or mutants by replacing serine 139 with alanine (S139A, a phospho-dead mutant) or glutamic acid (S139E, a phospho-mimetic form). An impaired Peli1 recruitment in H2AX-depleted cells was successfully rescued by either enforced expression of H2AX WT or H2AX S139E mutant. However, the expression of H2AX S139A mutant failed to rescue Peli1 recruitment at DSB sites in H2AX-depleted cells (Fig. 4g). We further confirmed that Peli1 interacted with H2AX S139E mutant through its FHA domain (Fig. 4h, i).

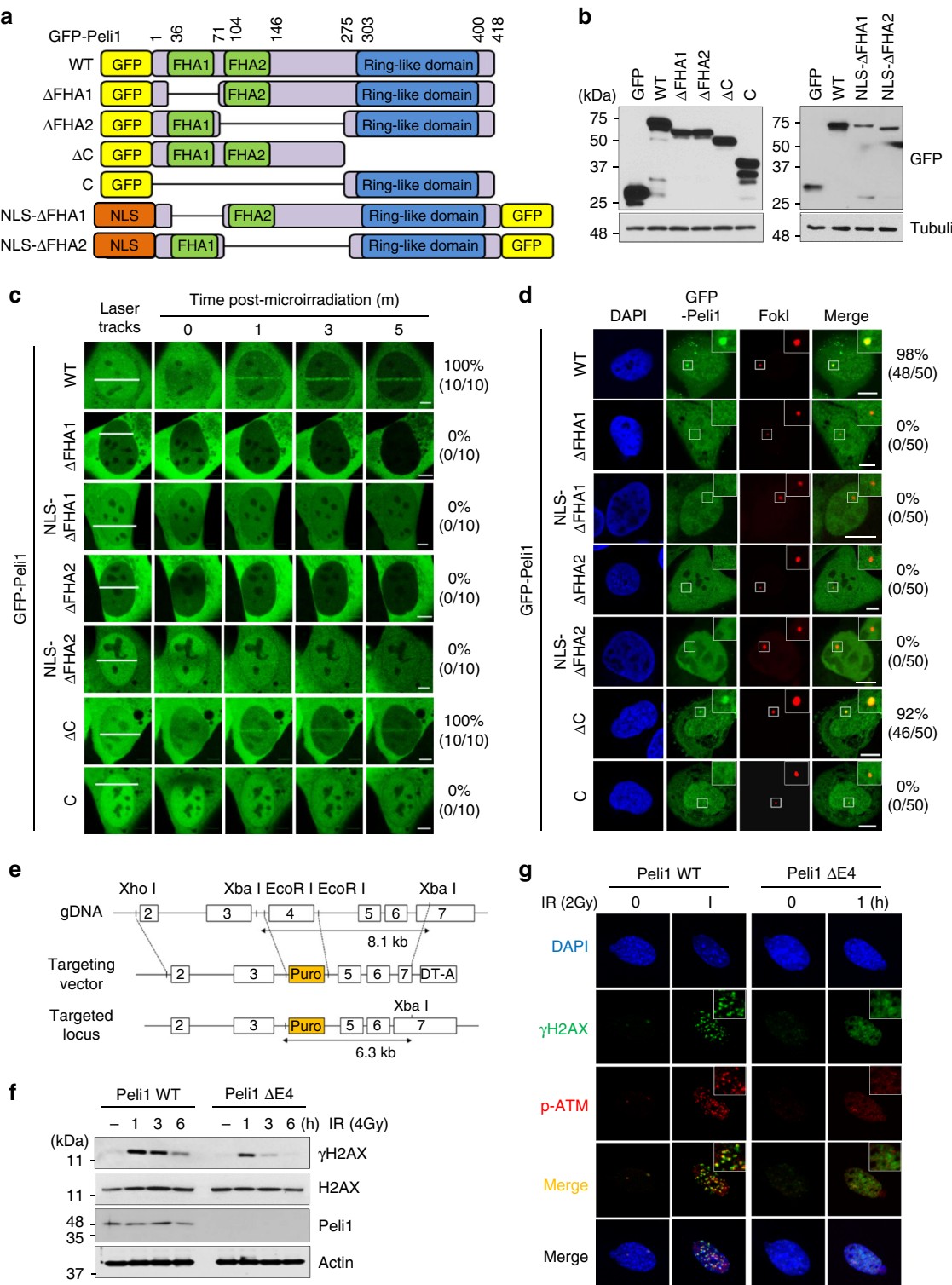

**Fig. 2** Deletion of Peli1 FHA domains impaired DNA damage response. **a** Diagrams of Peli1 WT, deletion, and NLS-fused deletion mutants. **b** 293T cells were transfected with plasmids encoding GFP-Peli1 WT, each of deletion and NLS-fused mutants. At 48 h post transfection, cells were collected and cell lysates were analyzed by immunoblotting with anti-GFP and anti-tubulin antibodies. **c** U2OS cells were transfected with GFP-Peli1 WT or each mutant and treated with BrdU (10 μM) for 30 h followed by laser micro-irradiation. Scale bar, 10 μM. **d** mCherry-LacI-FokI and GFP-fused Peli1 WT or mutants were cotransfected into U2OS-DSB reporter cells (U2OS 2–6–3). At 48 h, cells were fixed and visualized by confocal microscopy. Scale bars, 10 μm. **e** Targeting strategy to generate *PELI1* mutant mice in which coding for exon 4 was replaced with a puromycin cassette gene, resulting in Peli1 E4 truncated mice. **f** Peli1 WT and E4 truncated MEFs were treated with IR (4 Gy) as indicated. Cells were collected and then subjected to immunoblotting with Peli1, γH2AX, H2AX, and actin antibodies. **g** Peli1 WT and E4 truncated MEFs were treated with IR (2 Gy) and allowed to recover for 1 h before processing for γH2AX and phospho-ATM (p-Ser1981) immunostaining. Scale bars, 10 μm

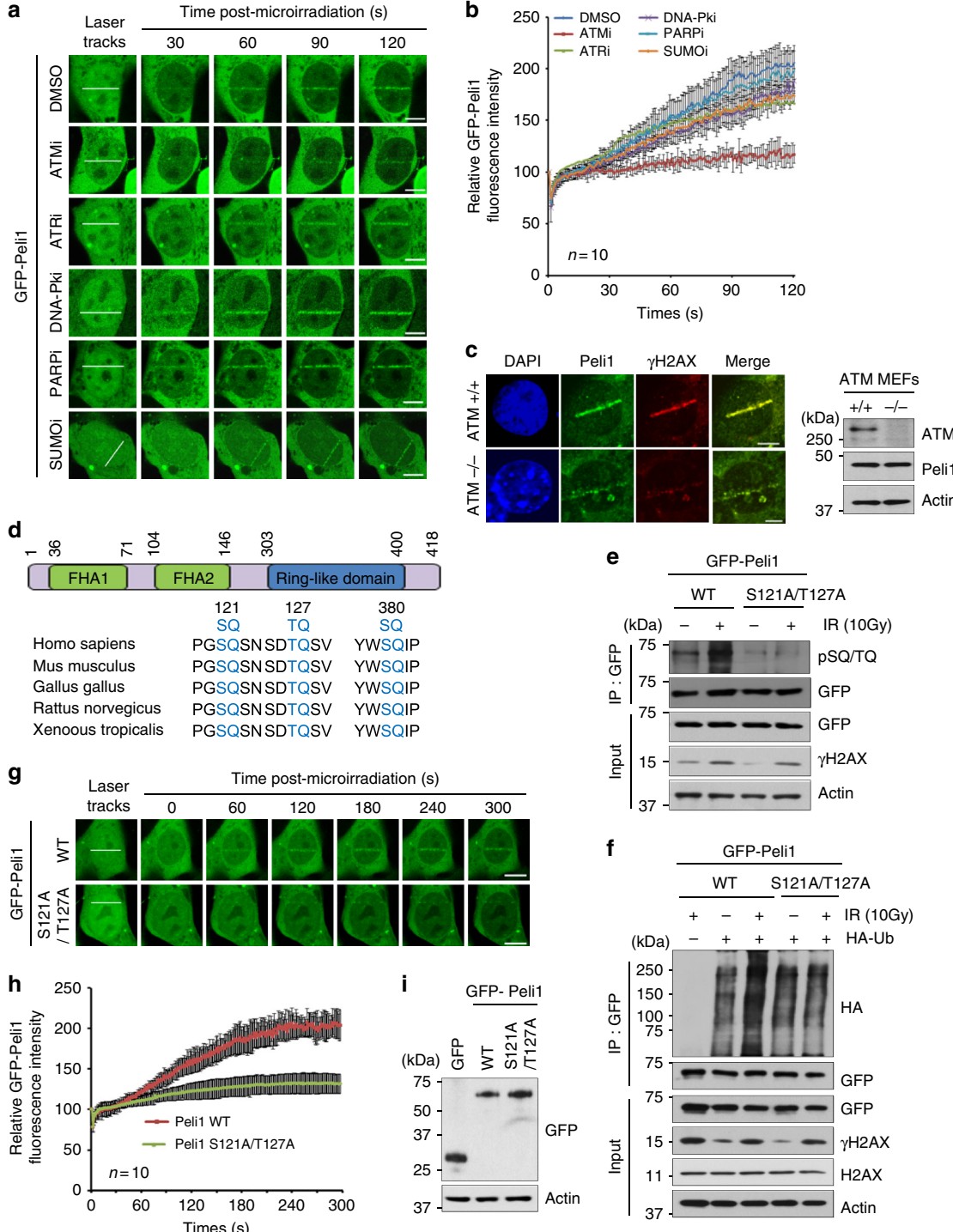

Collectively, these results suggest that the recruitment of Peli1 at DSB sites is dependent on the status of H2AX phosphorylation.

**Peli1-NBS1 interaction promotes ATM activation at DSB sites.** Recent studies have suggested that MRN complex plays an important role in the initial processing of DSB prior to repair by detecting damaged DNA and directly recruiting ATM to DNA foci for activation[36,37]. We thus asked whether Peli1 is required for DSB-induced ATM activation and recruitment to the DNA lesion. Interestingly, Peli1 ΔE4 MEFs showed significant reduction of ATM activation as detected by phosphorylated ATM S1981 antibody. It also resulted in reduced ATM recruitment to

laser stripes compared with Peli1 WT MEFs (Fig. 5a, b). Strikingly, impaired ATM activation and recruitment to DNA damage sites in Peli1-depleted cells were successfully rescued by enforced expression of Myc-Peli1 WT (pBabe-Peli1) compared with those of the control (pBabe) (Fig. 5c, d). We also confirmed that reintroduction of RFP-Peli1 WT, but not of RFP-Peli1 ΔE4, rescued the recruitment of phospho-ATM (p-ATM) and γH2AX at laser stripes in endogenous Peli1-depleted U2OS cells (Supplementary Fig. 6a, b). Notably, overexpression of Peli1 enhanced ATM activity (Supplementary Fig. 6c). Together, these results indicate that Peli1 promotes ATM activation and recruitment to DNA damage sites.

**Fig. 3** Peli1 is phosphorylated by ATM that translocate to DNA damage sites. **a, b** U2OS cells transiently expressing GFP-Peli1 were pre-treated with ATM (KU55933, ATMi, 10 μM), ATR (VE-831, ATRi, 10 μM), DNA-PK (KU57788, DNA-PKi, 10 μM), PARP (PJ34, PARPi, 100 μM), and SUMO (2Dd-08, SUMOi, 50 μM) inhibitors for 1 h. Cells were subjected to laser micro-irradiation. Laser stripes were examined at the indicated time point (**a**). Scale bars, 10 μm. The intensity of each laser stripe was determined by averaging values from ten cells at each time point and graphed (**b**). **c** ATM WT and KO MEF cells were subjected to laser micro-irradiation. Co-immunostaining with γH2AX and Peli1 antibodies at laser-induced DNA lesions (10 min after laser micro-irradiation). Scale bar, 10 μM. **d** Schematic showing potential Peli1 phosphorylation sites by ATM/ATR kinases. **e** 293T cells were cotransfected with GFP-Peli1 WT and S121A/T127A mutant. At 48 h post transfection, cells were treated with IR (10 Gy). At 30 min after IR, cells were collected and immunoprecipitated with an anti-GFP antibody. GFP-Peli1 protein complexes were subjected to immunoblotting with anti-pSQ/TQ, anti-GFP, anti-γH2AX, and anti-actin antibodies. **f** 293T cells were cotransfected with GFP-Peli1 WT and S121A/T127A mutant with or without HA-Ub. At 36 h post transfection, cells were treated with IR (10 Gy) and 30 min later cells were collected and immunoprecipitated with an anti-GFP antibody. GFP-Peli1 protein complexes were subjected to immunoblotting with indicated antibodies. **g, h** U2OS cells were transfected with plasmids encoding GFP-Peli1 WT or S121A/T127A mutant. At 48 h, cells were subjected to laser micro-irradiation and laser stripes were examined at indicated time point. Scale bar, 10 μM (**g**). The intensity of each laser stripe was determined by averaging values from ten cells at each time point and graphed (**h**). **i** U2OS cells were transfected with plasmids encoding GFP-Peli1 WT or S121A/T127A mutant. At 48 h post transfection, cells were collected and analyzed by immunoblotting with anti-GFP and anti-actin antibodies

We next hypothesized that Peli1 might interact with MRN complex and its upstream kinases. Extracts from asynchronously growing TAP and TAP-Peli1 (Flag-tagged Peli1)-transfected 293T cells were subjected to pull-down assays. It revealed that Peli1 strongly interacts with MRN complex, ATM, and CtIP, but not with ATR (Fig. 5e and Supplementary Fig. 6d). Their interactions were further augmented when cells were exposed to IR. Previously, it has been shown that recruitment of ATM to DSB sites is dependent on the MRN complex through ATM-NBS1 interaction[38,39]. Our immunoprecipitation assay also confirmed that Peli1 forms a complex with NBS1 in vivo (Fig. 5f). Peli1 usually binds to its protein substrates through FHA domain[30]. Thus, we asked whether Peli1 and NBS1 interaction mediates via the FHA domain. GFP-Peli1 WT and -FHA1 and -ΔC mutants readily interacted with FLAG-NBS1, GFP-Peli1 FHA2, but FLAG-C mutant failed to do so (Supplementary Fig. 7a, b). This indicates that the FHA2 domain of Peli1 is essential for NBS1 interaction. To identify domains of NBS1 responsible for Peli1 binding, we generated a series of deletion mutants of NBS1 in the context of Flag-tagged NBS1. WT, ΔFHA, ΔBRCT1, ΔBRCT2, ΔMA, and ΔC were cotransfected with GFP-tagged Peli1 WT followed by co-immunoprecipitation (Supplementary Fig. 7c, d). Although FLAG-NBS1 WT, -ΔFHA, -ΔBRCT2, -ΔMA, and -ΔC mutants localized at the nucleus (Supplementary Fig. 7e) and interacted with Peli1, FLAG-NBS1 ΔBRCT1 mutant failed to do so. This indicates that the BRCT1 domain of NBS1 is required for Peli1 interaction. Collectively, these results suggest that Peli1 promotes DSB-induced ATM activation and recruitment to DNA damage sites through NBS1 interaction.

**Peli1 phosphorylation by ATM promotes NBS1 ubiquitination.** Recent studies have shown that NBS1 ubiquitination promotes DSBs repair by HR via ATM activation and MRE11 recruitment[40,41]. To unveil whether Peli1 can ubiquitinate NBS1, 293T cells were transfected with GFP-Peli1 full length (FL), GFP-Peli1 ΔC, or GFP-Peli1 C mutant in combination with HA-Ub and Myc-NBS1, followed by immunoprecipitation with an anti-Myc antibody and subsequent immunoblotting. Interestingly, polyubiquitinated forms of NBS1 were evident in cells over-expressing Peli1 FL, but not in cells overexpressing Peli1 ΔC or Peli1 C mutant (Fig. 6a), suggesting that Peli1 FL increases the NBS1 polyubiquitination. Next, we doubted whether phosphorylation of Peli1 by ATM is necessary for NBS1 ubiquitination. To test this, in vivo ubiquitination assay was carried out using immunoprecipitated NBS1 from the cells transfected with GFP-Peli1 WT, two RING mutants of Peli1 [H313A (HA) and C336A (CA)][42], and phospho-dead mutant [S121A/T127A (AA)] in

combination with HA-Ub (Fig. 6b, c). Interestingly, levels of autoubiquitinated Peli1 in cells transfected with GFP-Peli1 WT were significantly increased by IR, whereas those in cells transfected with GFP-Peli1 RING mutants or phospho-dead mutant did not, suggesting that autoubiquitination of Peli1 dependens on its E3 ligase activity and ATM-mediated phosphorylation in DDR. Next, we performed in vivo ubiquitination assays using immunoprecipitated NBS1 from cells transfected with Peli1-targeting small interfering RNA (siRNA) (Fig. 6d) and GFP-Peli1 WT, and RING mutants, or a phospho-dead mutant (Fig. 6e). The ubiquitination of endogenous NBS1 was increased by IR and further augmented by Peli1 WT expression, whereas the depletion of Peli1 or the expression of Peli1 RING mutants or the phospho-dead mutant failed to ubiquitinate NBS1 (Fig. 6d, e). Furthermore, K63-linked ubiquitination of NBS1 was also augmented by Peli1 WT, but not Peli1 RING mutants or the phospho-dead mutant (Fig. 6e). Notably, K63-linked ubiquitination is known to provide a docking site for protein–protein interaction[43,44], whereas UBC13 is a major E2 (ubiquitin-conjugating enzyme) that triggers K63-linked ubiquitination[44,45]. Importantly, Peli1 seems to mediate the formation of K63-linked polyubiquitin chains onto NBS1. In the presence of UBC13, K63-linked ubiquitination of NBS1 was strongly induced by Peli1 expression. However, this signal was significantly attenuated by depletion of UBC13 (Fig. 6f). These results indicate that Peli1 mediates K63-linked ubiquitination of NBS1 in vivo.

Next, to predict Peli1-mediated ubiquitination sites of NBS1 in DDR, we examined the recruitment of FLAG-NBS1 WT and truncated mutants (ΔFHA, ΔBRCT1, ΔBRCT2, and ΔMA) at DSB sites. FLAG-NBS1 WT was strongly recruited to DSB sites, whereas NBS1 ΔFHA, ΔBRCT1, and ΔBRCT2 mutants showed reduced recruitment to DBS site (Fig. 7a) as reported previously[46]. Interestingly, deletion of MRE11- and ATM-interacting motif in NBS1 (ΔMA) also significantly diminished its recruitment to DSB sites. Next, we compared conserved lysine (K) residues in MA motif of NBS1 and found that K683, K686, and K690 residues of NBS1 are strongly conserved in eukaryotes (Fig. 7b). K665 and K683 residues of NBS1 have also been identified as ubiquitination substrate sites in human[47,48]. We thus tested whether these residues of NBS1 are involved in Peli1-mediated ubiquitination. NBS1 WT and K665/683R mutant retained the ability of K63-linked ubiquitination by Peli1, whereas NBS1 K686/690R mutant was unable to be ubiquitinated by Peli1 (Fig. 7c). Thus, K686 and K690 residues of NBS1 are indispensable for Peli1-mediated ubiquitination. We further tested whether Peli1-mediated NBS1 ubiquitination is required for ATM activation at DSB sites. NBS1 WT and K665/K683R mutant were recruited to and retained phospho-ATM at DSB

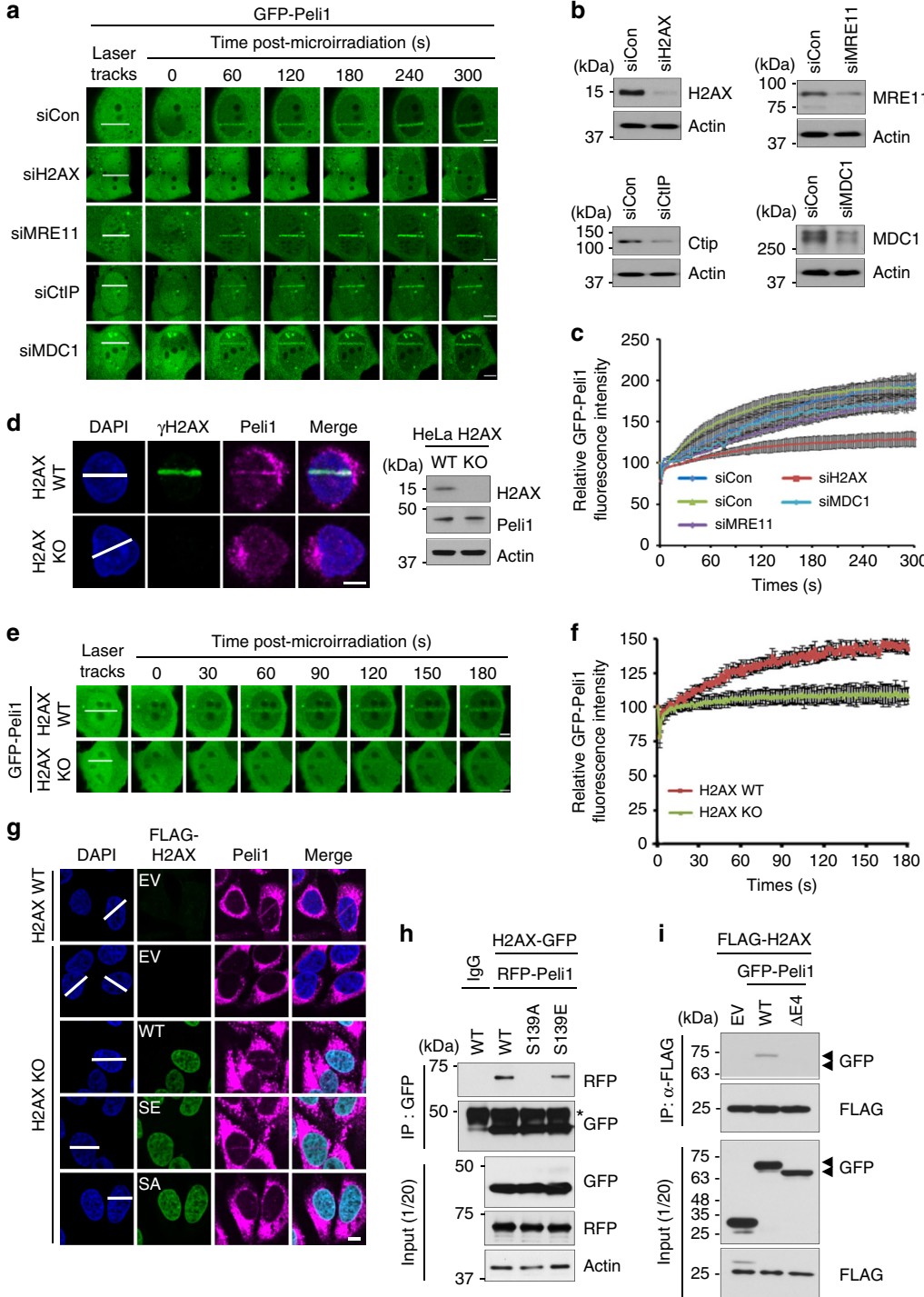

sites, whereas NBS1 K686/K690R mutant failed to do so (Fig. 7d). Consistent with the change of p-ATM level, protein amounts of GFP-NBS1 K686/690R mutant were also significantly reduced in chromatin fractions, but not in whole-cell lysate, compared with those of GFP-NBS1 WT and K665/683R mutant (Fig. 7e). We further tested whether the Peli1 ligase activity and ATM-mediated phosphorylation are required for feedback activation of ATM at DSB sites. Only Peli1 WT, but not Peli1 RING mutants or phospho-dead mutant, rescued p-ATM signal at DSB sites (Supplemenatry Fig. 8a, b).

To better understand how Peli1 regulates ATM activation upon DNA damage, we examined whether Peli1 regulates the interaction between ATM and NBS1. Interestingly, the

interaction of NBS1 with ATM was impaired in Peli1-depleted cells compared with that in controls (Fig. 8a), consistent with defects of ATM recruitment to DSB sites (Fig. 5a, b, d). We next asked whether Peli1 affects stabilities of MRE11 and NBS1. Peli1 ΔE4 MEFs showed a significant reduction in the level of MRN complexes compared with Peli1 WT MEFs (Fig. 8b). Moreover, treatment with proteasome inhibitor (MG132) increased levels of MRN complex in both Peli1 WT and Peli1 ΔE4 MEFs (Fig. 8c), suggesting that the stability of MRN complexes could be regulated by Peli1. However, contrary to the above experiment, ectopic expression of Peli1 augmented levels of NBS1 expression in a dose-dependent manner, but not those of MRE11 or RAD50 (Fig. 8d). Next, our experiment by ectopic expression of Peli1 WT

**Fig. 4** γH2AX-mediated accumulation of Peli1 at DNA damage sites. **a**, **b** Mobilization kinetics of GFP-Peli1 to sites of DNA damage. U2OS cells were cotransfected with plasmids encoding GFP-Peli1 and indicated siRNAs. After 48 h, cells were subjected to laser micro-irradiation and laser stripes were examined at indicated time point. Scale bar, 10 μM. U2OS cells were cotransfected with plasmids encoding GFP-Peli1 and indicated siRNAs (**a**). After 48 h, cell lysates were analyzed by immunoblotting with indicated antibodies (**b**). **c** The intensity of each laser stripe was determined by averaging values from ten cells at each time point and graphed. **d** H2AX WT and KO HeLa cells were subjected to laser micro-irradiation. Colocalization of Peli1 and γH2AX at laser-induced DNA lesions (10 min after laser micro-irradiation; left panels) and western blotting analysis (right panels) are shown. Scale bar, 10 μM. **e**, **f** H2AX WT and KO HeLa cells were transfected with GFP-Peli1. At 48 h, cells were subjected to laser micro-irradiation. Laser stripes were examined at the indicated time point (**e**). The intensity of each laser stripe was determined by averaging values from ten cells at each time point and graphed (**f**). **g** H2AX WT and KO HeLa cells were transfected with FLAG empty vector (EV), FLAG-H2AX WT, S139A, or S139E mutant. At 48 h post transfection, cells were subjected to laser micro-irradiation. After 10 min post micro-irradiation, cells were fixed and immunostained with indicated antibodies. Scale bar, 10 μM. **h** H2AX HeLa cells were cotransfected with H2AX-GFP WT, S139A, or S139E mutant. After 48 h, cells were collected and immunoprecipitated with an anti-GFP antibody. These protein complexes were subjected to immunoblotting with indicated antibodies. **i** 293T cells were cotransfected with FLAG-H2AX WT and GFP empty vector (EV), GFP-Peli1 WT, or GFP-Peli1 E4 truncated. At 48 h post transfection, cells were collected and immunoprecipitated with an anti-FLAG antibody. These protein complexes were subjected to immunoblotting with antibodies shown

or mutants in endogenous Peli1-depleted cells further confirmed that the depletion of Peli1 led to the complete loss of NBS1 recruitment to laser irradiation-induced DSB sites, whereas the continuous expression of Peli1 maintained the recruitment of NBS1 at DSB sites (Fig. 8e). Similarly, Peli1 ΔE4 MEF cells showed significant reduction in NBS1 recruitment to DSB sites compared with Peli1 WT MEF cells (Fig. 8f and Supplementary Fig. 9a, b). However, depletion of NBS1 did not affect the ability of Peli1 to accumulate at DSB sites (Supplementary Fig. 9c–e), whereas recruitment of MRE11 and RAD50 to DSB sites was inhibited by Peli1 depletion (Supplementary Fig. 10). Taken together, these results indicate that Peli1 is required for ATM activation through NBS1 ubiquitination.

**Peli1 is required for DNA-end resection-mediated HR repair**. The major event in DSB repair pathway is DNA-end resection for NHEJ or HR repair[3,35]. To understand how Peli1 involves in DSB resection, we examined whether Peli1 depletion affects recruitment of various factors known to be involved in DNA-end resection. Protein levels of CtIP, Exo1, p-BRCA1, RPA32, or RAD51 did not alter in both Peli1 WT and ΔE4 MEFs (Fig. 9a). However, recruitments of CtIP, p-BRCA1, Exo1, RPA32, and Rad51 to laser stripes were markedly reduced in Peli1 ΔE4 MEFs (Fig. 9b, c). Moreover, Peli1 depletion impaired recruitments of MDC1, 53BP1, and FK2 to laser-damaged sites (Supplementary Fig. 11a–c). These results indicate that Peli1 contributes to DNA-end resection and HR repair downstream of the MRN complex.

To further evaluate biological functions of Peli1 in DNA damage repair, especially HR and NHEJ, we took advantage of stably or plasmid-based expressing GFP reporter systems. Interestingly, we found that Peli1-depleted cells showed reduction of HR-repaired GFP-positive population compared with Peli1 WT cells, suggesting defects of HR repair by Peli1 depletion (Fig. 9d and Supplementary Fig. 12a–c). However, there was no difference in NHEJ repair competence between Peli1 WT and Peli1-depleted cells (Supplementary Fig. 12d–f). To further determine whether depletion of Peli1 affects HR efficiency simply by altering cell cycle distribution, we examined cell cycle profile of Peli1-depleted cells generated by small hairpin RNA (shRNA) or siRNA-targeting Peli1 (shPeli1 and siPeli1 3′-untranslated region (UTR)). Peli1 depletion did not alter cell cycle progression (Supplementary Fig. 12g, h). Next, we tested whether Myc-Peli1 WT, RING mutants, and phospho-dead mutant could rescue HR repair. Ectopic expression of Peli1 WT recovered HR repair, whereas Peli1 RING or phospho-dead mutant was dispensable for rescue HR repair in endogenous Peli1-depleted cells (Fig. 9e and Supplementary Fig. 12i). Further, we confirmed the rescue experiment by adding of RFP-Peli1 WT and ΔE4 constructs into endogenous Peli1-

depleted cells (Fig. 9f). Collectively, these results indicate that Peli1 plays a critical role in regulating HR repair rather than NHEJ. To further evaluate biological functions of Peli1 in DDR, we performed a clonogenic assay in two different types of Peli1-depleted cells (by shPeli1 or 3′-UTR shPeli1 transfection) following IR. Peli1 ΔE4 MEFs displayed much higher sensitivity to IR than Peli1 WT MEFs (Fig. 9g). Similarly, Peli1-depleted U2OS cells (by shPeli1 or 3′-UTR shPeli1 transfection) also showed increased IR sensitivity (Fig. 9h). Collectively, these results demonstrate that Peli1 E3 ligase contributes to DNA-end resection-mediated HR repair

## Discussion

DNA damages are recognized by proteins to trigger and coordinate the recruitment and activation of additional DNA repair proteins, which elicit cascades of posttranslational modifications, including phosphorylation, ubiquitination, and sumoylation that orchestrate DNA damage signaling and repair. These events are largely initiated by apical DDR kinases ATM and ATR. Here we proposed a direct association between ATM and Peli1 in coordinating pathways of DNA damage recognition, signaling, and repair. Our findings indicate that Peli1 is activated by ATM-mediated phosphorylation and recruited to sites of DNA damage in ATM- and γH2AX-dependent manners. In addition, Peli1 promotes effective DNA-end resection, a process required for MRN complex signaling from DSB sites and DSB repair by HR. Specifically, Peli1 is likely to be an early DSB signal-responsive ubiquitin ligase, because Peli1 is activated by ATM-mediated phosphorylation and recruited to DSB sites at a time similar to another initial DDR protein NBS1. Reversely, Peli1 is required for ATM activation at DSB sites, indicating that an interaction between ATM and Peli1 is essential for activating ATM via MRN complex (Fig. 5 and Supplementary Fig. 6).

Phosphorylation of H2AX (γH2AX) is one of hallmarkers upon DNA damage. Roles of other histone modifications such as acetylation, methylation, and ubiquitination have also been recently clarified, particularly in the context of HR repair. Interestingly, our results show that ATM-mediated phosphorylation allows Peli1 links to γH2AX. This process relays further important processes for interaction with NBS1, followed subsequent ubiquitination of NBS1. Therefore, Peli1 phosphorylation is critical for regulation of cellular DDR through NBS1 ubiquitination. However, it is unclear whether the interaction of Peli1 with γH2AX contributes to HR-related histone modification and chromatin remodeling.

As an initiation of DDR requires DDR-related protein complexes to access and accumulate at DNA damage sites, protein–protein interactions play important roles in DDR. Peli1 FHA domains share very high sequence identities[30,31]. Peli1 is

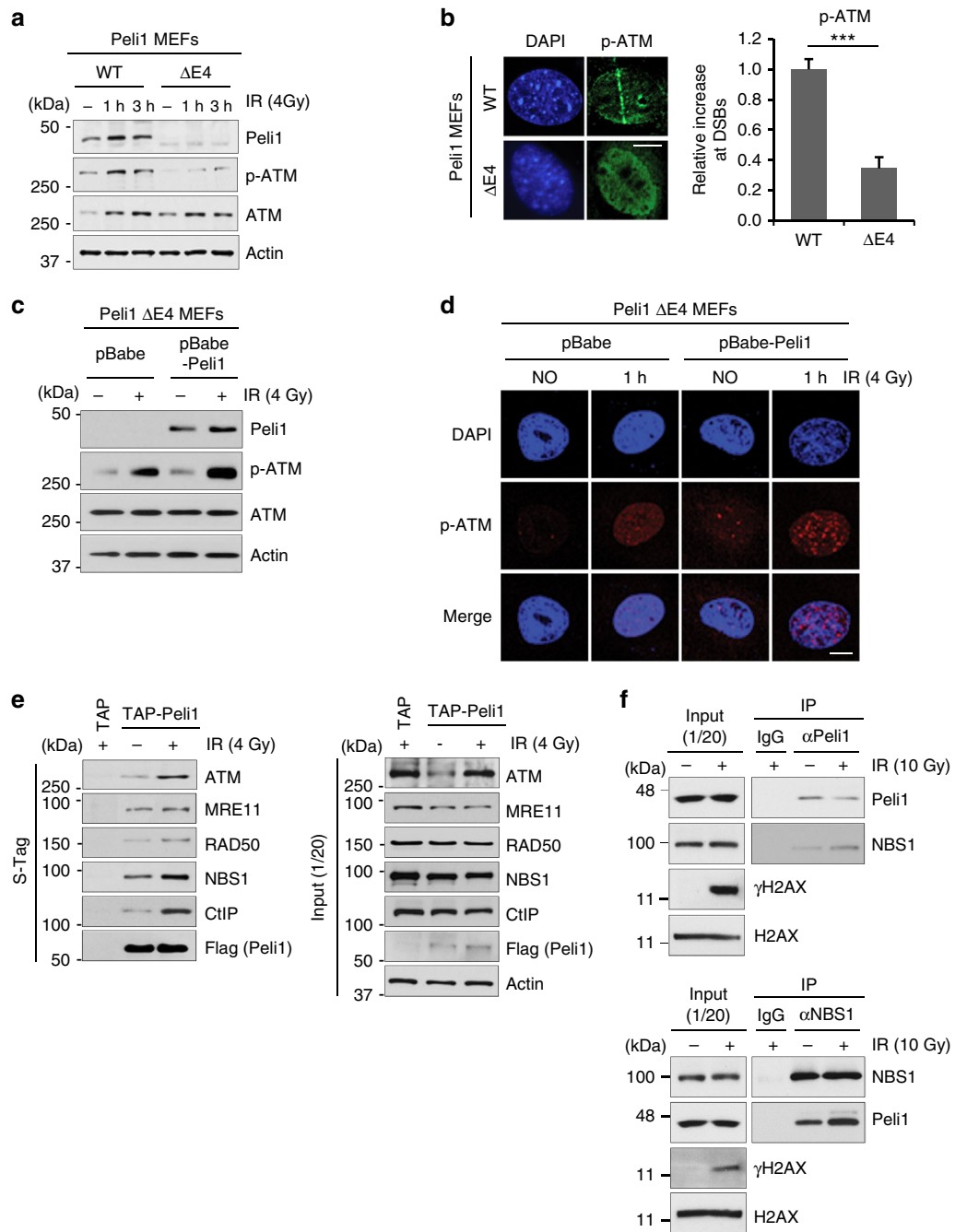

**Fig. 5** Peli1-mediated ATM activation and NBS1 interaction upon DNA damage. **a** Peli1 WT and ΔE4 MEF cells were treated with 4 Gy IR. At indicated time, cells were collected and then subjected to immunoblotting. **b** Peli1 WT and E4 truncated MEF cells were micro-irradiated, fixed after 10 min, and immunostained with anti-p-ATM antibody (left panels). Mean levels of p-ATM accumulation at sites of laser tracks were quantified using Image J software and plotted as indicated (right panels). Data show mean ± SEM; n = 20 cells. Student's t-test was used for statistical analyses. Scale bars, 10 μm. **c** Peli1 E4 truncated MEFs were transfected with a control pBabe or pBabe-Peli1 (Myc-Peli1) retrovirus and treated with IR (4 Gy). At 1 h post-IR, cells were collected and immunoblotted with anti-Peli1, anti-p-ATM, anti-ATM, and anti-actin antibodies. **d** Peli1 E4-truncated MEFs were transfected with pBabe or pBabe-Peli1 (Myc-Peli1) retrovirus for 24 h and then treated with IR (4 Gy). At 1 h post-IR, cells were fixed and immunostained with anti-p-ATM antibodies. Scale bars, 10 μm. **e** 293T cells were transfected with TAP (control) or TAP-Peli1 (Flag-tagged Peli1). At 36 h, cells were treated with or without IR (4 Gy). At 30 min post-IR, cells were collected and isolated through S-tag pull-down assay. Bound proteins were immunoblotted with indicated antibodies. **f** IR-treated or non-treated 293T cell lysates were immunoprecipitated with anti-NBS1 or anti-Peli1 antibody and immunoblotted with antibodies indicated. γH2AX was used as a positive marker of IR in input panel

rapidly recruited to and accumulated at damaged sites. However, deletion of its FHA domain leads to loss of recruitment to DSB sites. Interestingly, FHA domains are present on a wide range of DDR-related proteins such as Chk2 and NBS1. These FHA domains are important for integrating upstream signals[31]. In addition, DDR-related proteins having potential phosphorylation sites are targets for FHA domains of ubiquitin ligases[49]. Taken together, these findings suggest that Peli1 has a scaffold function that facilitates complex formation necessary for DNA-damage signaling networks.

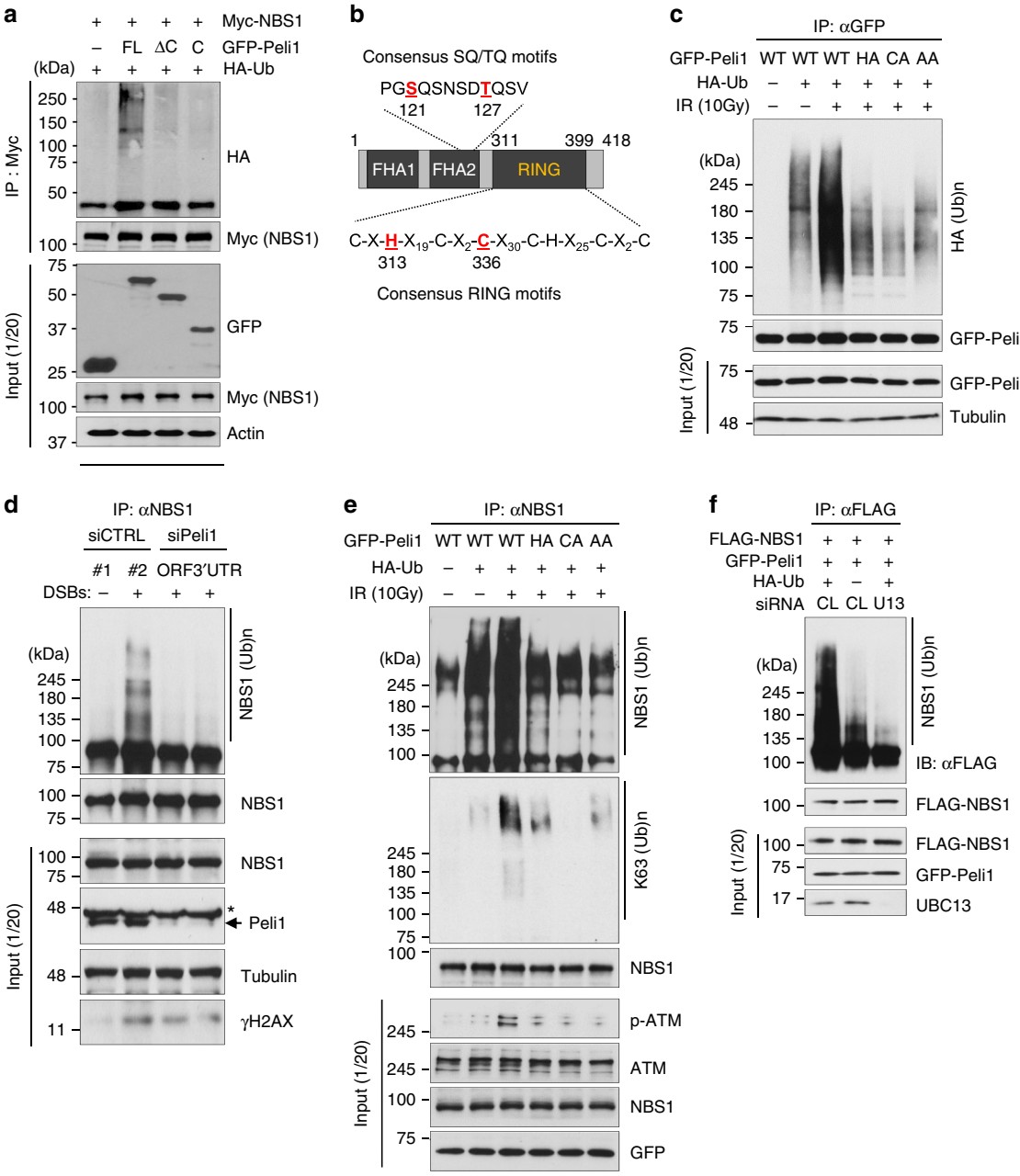

**Fig. 6** Phosphorylation-mediated Peli1 activates NBS1 ubiquitination in vivo. **a** 293T cells were transfected with GFP-Peli1 WT, GFP-Peli1 C-terminal deletion, or C-terminal mutant in combination with HA-Ub and Myc-NBS1. At 36 h post transfection, cells were collected and immunoprecipitated with an anti-Myc antibody. NBS1 protein complexes were subjected to immunoblotting with anti-HA and anti-Myc antibodies. **b** Diagram of consensus motifs of SQ/TQ and RING domain of Peli1 protein in eukaryotes. **c** 293T cells were transfected with GFP-Peli1 WT, H313A (HA), C336A (CA), or S121A/T127A (AA) together with HA-Ub. At 48 h, cells were treated with IR. Cell lysates were immunoprecipitated with anti-GFP antibody. Autoubiquitination of GFP-Peli1 constructs was detected with anti-HA antibody. **d**, **e** K63-linked ubiquitination of NBS1 by Peli1. siRNA-targeting Peli1 (**d**) and plasmid encoding GFP-Peli1 (**e**) were transfected into 293T cells, respectively. Cell lysates were immunoprecipitated with anti-NBS1 antibody. NBS1 ubiquitination was detected by anti-NBS1-, anti-HA-, and K63-specific antibodies. **f** 293T cells were transfected with GFP-Peli1, FLAG-NBS1, and HA-Ub. At 24 h post transfection, siRNAs targeting UBC13 #1 and #2 (U13), or control (CL) was reintroduced into cells. After 48 h, cells were immunoprecipitated with anti-FLAG antibody. NBS1 ubiquitination was detected with anti-FLAG antibody

An importance of ubiquitination in DDR indicates by orchestrated recruitment of proteins such as 53BP1 and BRAC1 onto damaged chromatin[50]. These events are initiated by γH2AX, which is recognized by MDC1[19]. MDC1 is phosphorylated by ATM and bound to RNF8 through its FHA domain[20,51,52], then RNF8 targets ubiquitination of proteins at DSB sites. RNF168 also ubiquitinates histones at DSB sites and promotes the assembly of repair proteins[49,53]. Therefore, the primary outcome of RNF8/

RNF168-dependent ubiquitination is recruitment and retention of DSB signaling and repair factors on damaged chromatin. Although identification of ubiquitin-dependent cascades is important, none of the prior work has identified the enzymatic cross-talk involving ATM kinase that can directly regulate the activity of DSB-associated ubiquitin ligase and contribute to subsequent recognition of DSBs and the process of DSB repair. Similar to RNF8/RNF168 tandem, Peli1 recognizes γH2AX and

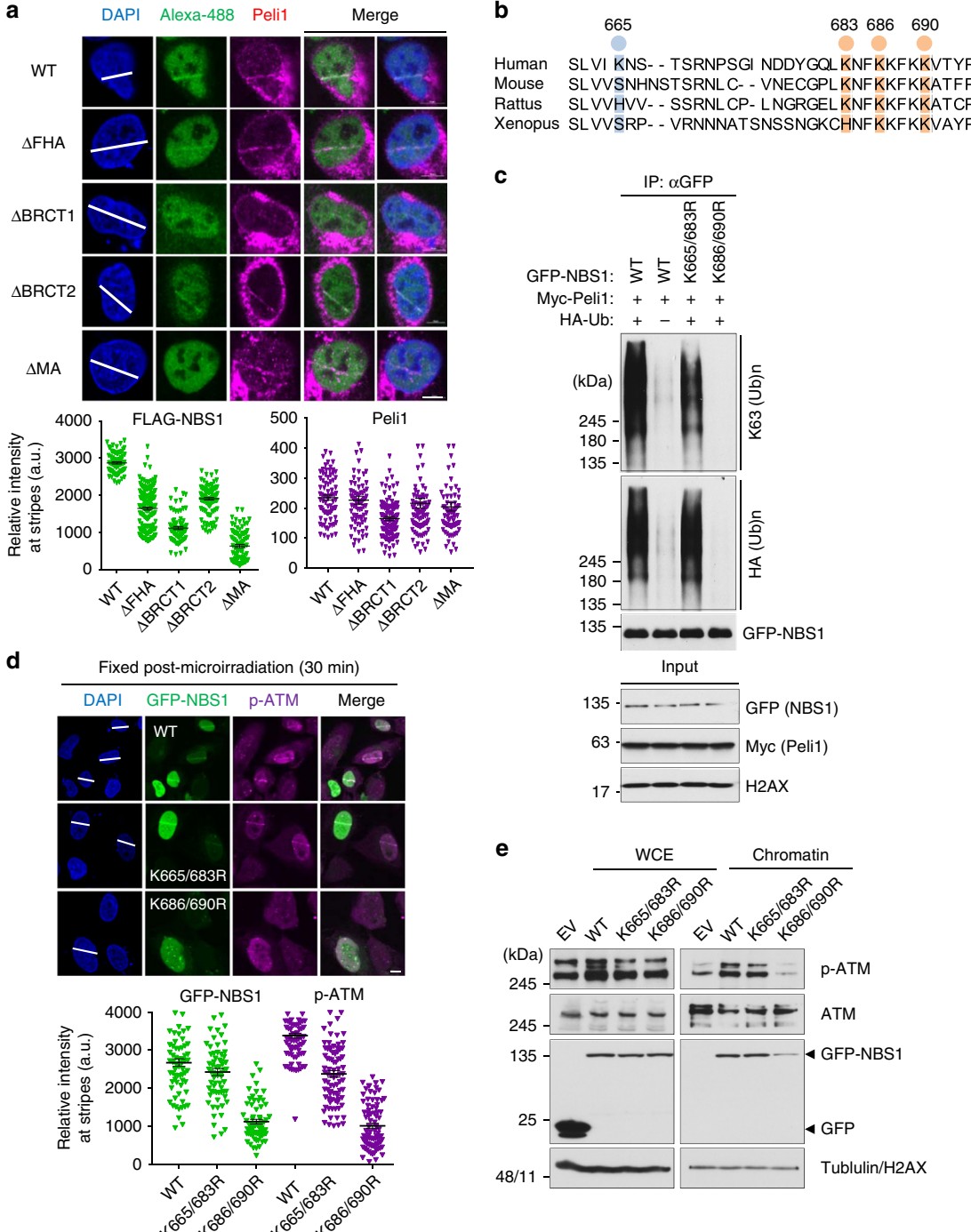

**Fig. 7** Peli1 ubiquitinates NBS1 at lysine 686 and 690 that regulate NBS1 and active ATM retention at damaged sites. **a** FLAG-NBS1 WT and truncated mutants were transfected into U2OS cells. At 48 h post transfections, cells were microirradated and fixed at 30 min post micro-irradiation. Recruitments of Peli1 and FLAG-NBS1 constructs were evaluated with anti-FLAG and anti-Peli1 antibodies (upper panels). Quantitative analysis of FLAG-NBS1 constructs and Peli1 at laser stripes was also performed. The intensity of each constructs at laser stripes was determined by averaging values from 20 cells and graphed (lower panels). Scale bars, 10 μm. **b** Comparison of conserved lysine residues of C-terminal NBS1 in eukaryotes. **c** 293T cells were transfected with GFP-NBS1 WT and lysine-dead (K665/683R and K686/690R) mutants in combination with Myc-Peli1 WT and HA-Ub. Immunoprecipitated NBS1 with anti-GFP antibody was detected with anti-HA and anti-K63 antibodies. **d** Cells transfected with GFP-Peli1 WT, K665/683R, and K686/690R mutants were subjected to micro-irradiation at 24 h post transfection with endogenous NBS1 3′-UTR-targeting siRNA. At 30 min post micro-irradiation, micro-irradiated cells were fixed and immunostained with anti-p-ATM antibody (upper panels). Quantitative analysis of GFP-NBS1 constructs and p-ATM at laser stripes was also performed. The intensity at laser stripes was determined by averaging values from 20 cells and graphed (lower panels). Scale bars, 10 μm. **e** 293T cells were transfected with GFP-NBS1 WT, K665/683R, or K686/690R mutants. At 48 h post transfection, DSB-mimetic drug phleomycin (50 μg/ml) was used to treat cells. After washing out, whole cell extracts and chromatin fraction were collected 30 min later. Indicated antibodies were used to detect signals

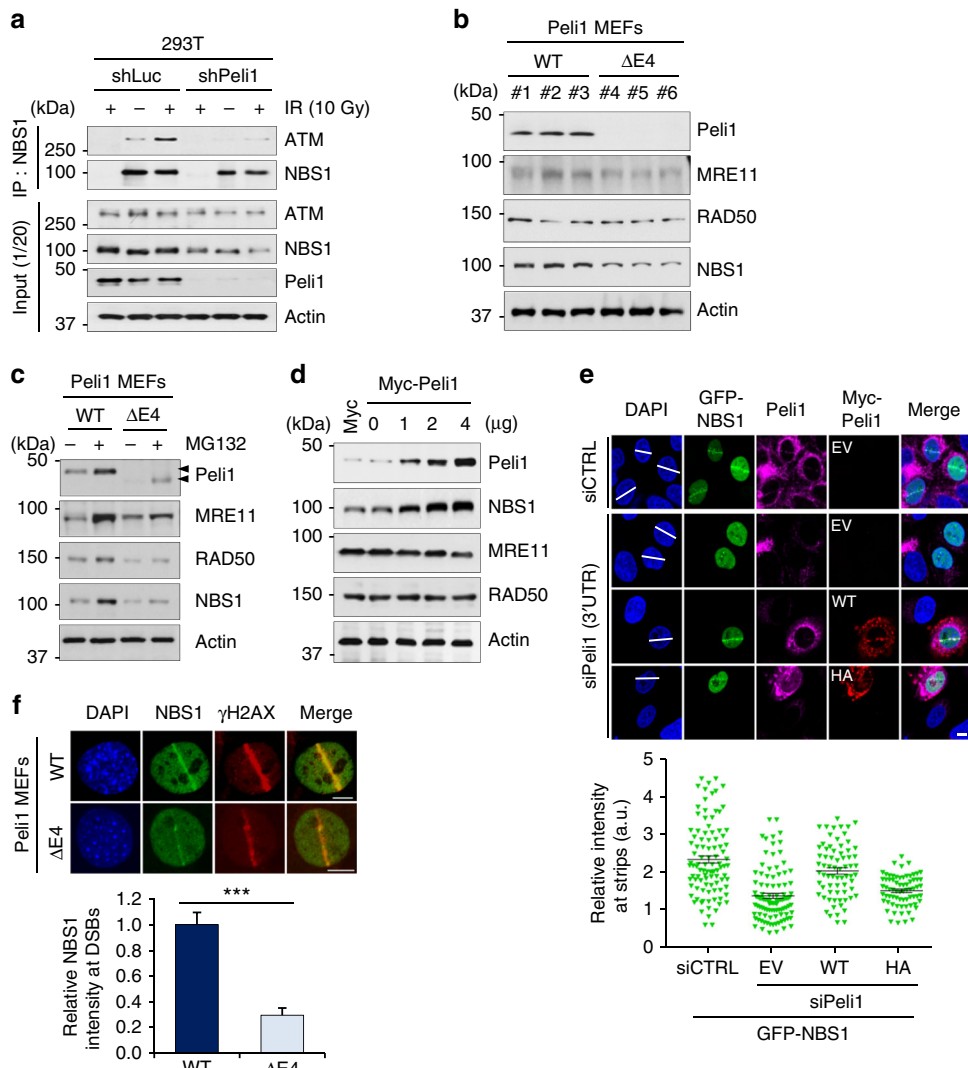

**Fig. 8** Peli1-mediated NBS1 ubiquitination stabilizes the Mre11, RAD50, and NBS1 (MRN) complex. **a** 293T cells were transfected with shLuc or Peli1 and treated with or without IR (10 Gy). At 30 min post-IR, cells were collected and immunoprecipitated with an anti-NBS1 antibody. NBS1 protein complexes were subjected to immunoblotting. **b** Immunoblotting of MRN complex components in Peli1 WT and E4 truncated MEF cells using antibodies indicated. **c** Peli1 WT and E4 truncated MEF cells were treated with proteasome inhibitor MG132 (10 μM) for 6 h. Cells were subsequently subjected to immunoblotting. **d** 293T cells were transfected with Myc-Peli1 expression plasmid. At 48 h post transfection, cells were collected and cell lysates were analyzed by immunoblotting with antibodies indicated. **e** siRNA-targeting control (CTRL) or endogenous Peli (3′-UTR) was transfected into U2OS cells in combination with GFP-NBS1. U2OS cells reintroduced with empty vector (EV), Myc-Peli1 WT, or RING mutant (HA) were subjected to micro-irradiation at 48 h. At 30 min post micro-irradiation, cells were fixed and immunostained with anti-Peli1 and anti-Myc antibodies. The intensity of GFP-NBS1 at laser stripes was determined by averaging values from 10 to 20 cells and graphed. Scale bars, 10 μm. **f** Peli1 WT and E4 truncated MEF cells were micro-irradiated with a UV laser, fixed after 10 min, and immunostained with anti-NBS1 and anti-γH2AX antibodies (upper panel). Scale bar, 10 μM. Mean levels of NBS1 accumulation at sites of laser tracks were quantified (lower panel). Student's *t*-test was used for statistical analyses

subsequently catalyzes K63-linked ubiquitination of NBS1, leading to further activation of ATM. An important question is how ubiquitination signal activates DSB signals. Our data indicate that Peli1 is directly activated by ATM-mediated phosphorylation, which then augments ATM activation via NBS1 ubiquitination. Although the RNF8 pathway can promote HR, RNF8/RNF168-dependent ubiquitination promotes NHEJ during immunoglobulin class switching and dysfunctional telomere fusion[54,55]. However, Peli1 contributes to HR repair, but not NHEJ. Therefore, it is likely to be that Peli1-mediated damage signaling is different from other types of ubiquitin ligase for the recognition of DSBs and DSB repair.

It seems that the amount of Peli1 is rate limiting for homeostatic regulation[25]. Peli1 expression is highly suppressed under normal or non-pathological conditions, whereas it is activated in response to various stress stimuli[28,29,56]. For instance, Peli1 expression is upregulated in patients with neutrophilic asthma[57] and those with diffuse large B-cell lymphoma[25]. Moreover, Peli1 expression is associated with poor outcomes in B-cell lymphomas[25], suggesting that aberrant expression of Peli1 is linked to the development of human diseases. Thus, unbalanced activation or expression of Peli1 can trigger forced deregulation of DSB signal cascades, ultimately contributing to defects in DNA repair and associated processes. The finding that ATM-mediated phosphorylation of Peli1 stimulates its ubiquitin ligase activity provides that the activity and stability of Peli1 are regulated by phosphorylation-dependent intracellular signal. Moreover, a functional role for Peli1 in DDR is highly distinct from its previously established roles in tumorigenesis and autoimmunity.

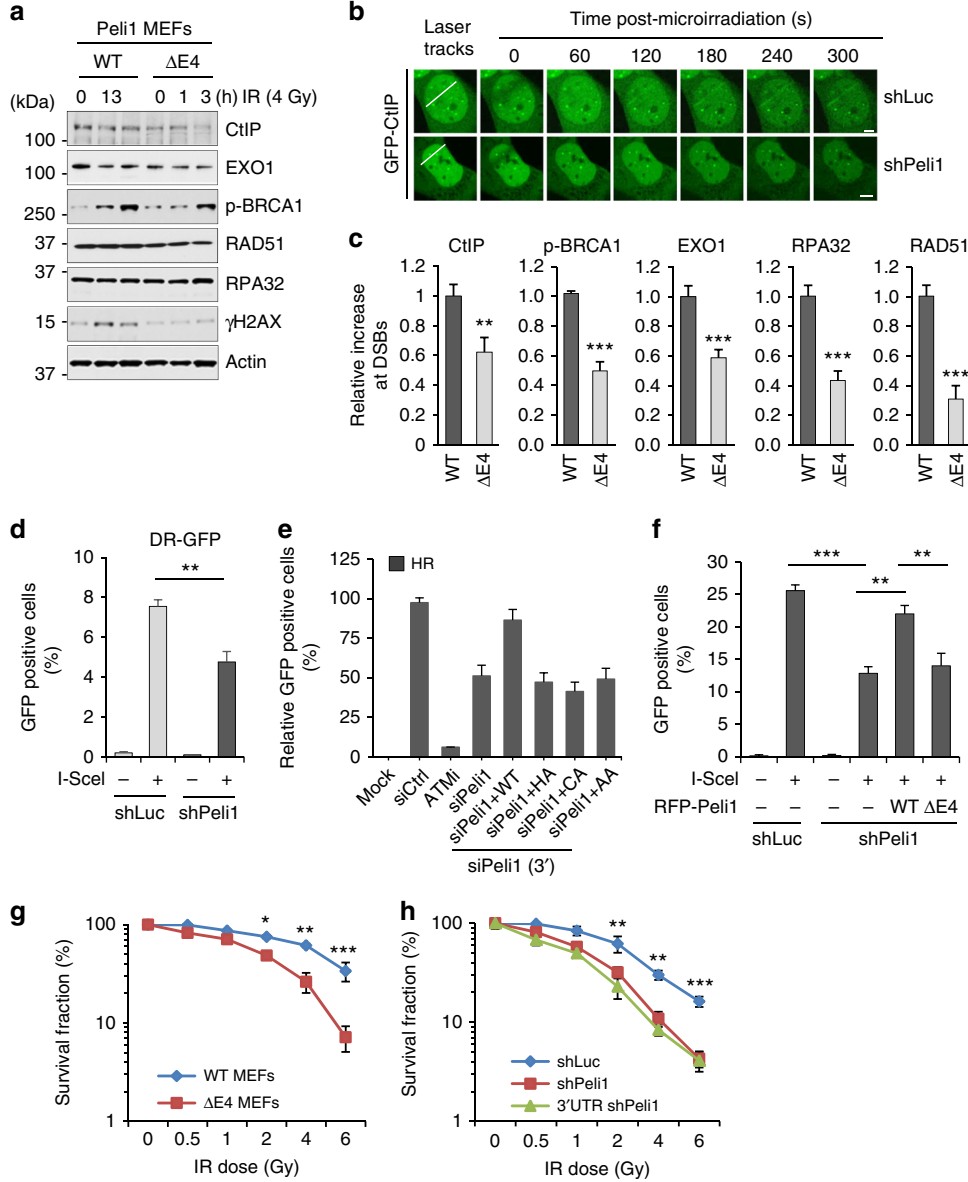

**Fig. 9** Peli1 involves DNA-end resection for HR repair. **a** Peli1 WT and E4 truncated MEF cells were treated with IR (4 Gy). At 0, 1, and 3 h post-IR, cells were collected and immunoblotted. **b** U2OS cells stably expressing GFP-CtIP were transfected with shLuc or shPeli1. After 48 h, cells were subjected to laser micro-irradiation. Laser stripes were examined at indicated time points. Scale bars, 10 μm. **c** Peli1 WT and E4 truncated MEF cells were micro-irradiated at 10 min for CtIP and p-BRCA1, at 30 min for Exo1, RPA32, and Rad51. Fixed cells were immunostained with indicated antibodies. All data are presented as mean values ± SEM of three independent experiments; $n = 20$ cells. Student's $t$-test was used for statistical analyses. **d, e** U2OS HR reporter cells were transfected with shRNA targeting Peli1 or control (**d**), or siRNA-targeting Peli1 3′-UTR region and Myc-Peli1 WT, RING mutants (HA and CA), and phospho-dead mutant (AA) in combination as indicated (**e**). After 24 h, I-SceI nuclease-expressing vector was transfected into reporter cells. After 48 h, GFP-positive cells were calculated by FACS. Student's $t$-test was used for statistical analyses. An ATM inhibitor (ATMi) was used as a HR positive control (**e**). **f** Twenty-four hours after transfection of shLuc or shPeli1 vector in 293T cells expressing pDR-GFP, I-SceI expression vector was further transfected into these cells to generate a DSB within Sce-GFP. FACS analysis was then performed to quantify HR-repaired GFP-positive cells. Student's $t$-test was used for statistical analyses. **g** Peli1 WT or E4 truncated MEF cells were treated with indicated doses of IR. Clonogenic survival assays were performed three independent experiments and survival curves were generated. Results are presented as mean ± SEM. Student's $t$-test was used for statistical analyses. **h** U2OS cells silenced with control (shLuc) or Peli1 (shPeli1 or 3′-UTR shPeli1) were treated with various doses of IR and survival rate of these cells was determined by colony formation assay. Results are mean ± SEM of three independent experiments. Student's $t$-test was used for statistical analyses

NBS1 is a multifunctional protein involved in various DDRs. Its recently identified binding partner, RNF20, is also a ubiquitin ligase that facilitates mono-ubiquitination of histone H2B, a process that is crucial for recruitment of chromatin remodeler SNF2h to DSB sites[58]. It is conceivable that NBS1 is a component of the MRN complex that participates in the HR pathway, whereas its role in NHEJ repair is much more limited[10]. Upon MRN complex recruitment to DSB sites, NBS1/RNF20-dependent H2B ubiquitination and SNF2h-dependent chromatin remodeling can stimulate DNA-end resection by MRN/CtIP and other nucleases[10]. Moreover, NBS1 is essential for ATM activation. Here we demonstrate that Peli1 facilitates K63-linked

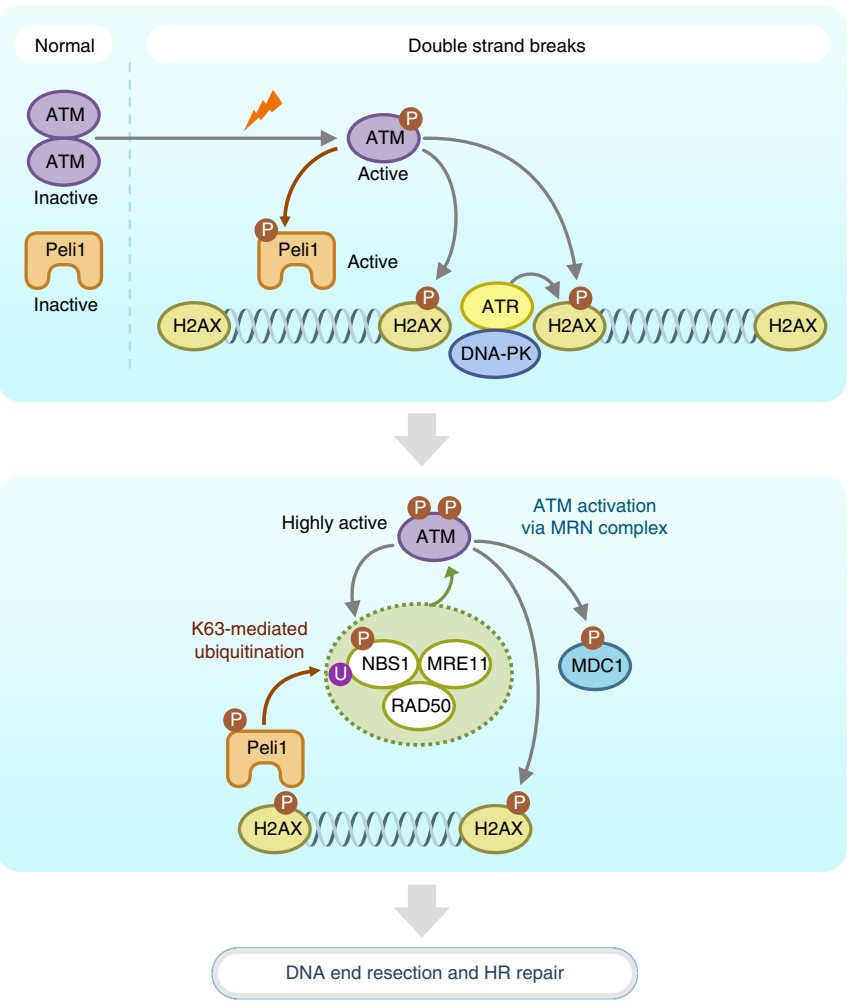

**Fig. 10** Model of Peli1-mediated DNA damage signaling and HR repair. Peli1 recruitment to DSB sites is triggered by ATM-mediated phosphorylation and directly binds to phosphorylated histone H2AX (Step I). Once recruited, Peli1 interacts with NBS1 and catalyzes K63-linked ubiquitination of NBS1, leading to further activation of ATM, which reinforces DNA-end resection and HR repair (Step II)

ubiquitination of NBS1, which further promotes ATM activation, suggesting that DSB-induced ATM activation is in part dependent on Peli1 via NBS1 ubiquitination. Accumulated evidence indicates that DNA-end resection is a two-stage process, with initially limited resection due to MRN/CtIP involvement and subsequent longer-range processing by mechanisms involving EXO1 and BLB[59,60]. Our findings show that Peli1 as an ubiquitin ligase for NBS1 is likely to contribute to this key transition between short- and long-term repair in human and mice cells.

In contrast with DSB-induced phosphorylation, other post-translational mechanisms for maintaining and retaining ATM activation during DSB response and HR repair are less clear. We propose that Peli1 is an additional DDR factor that contributes to ATM and γH2AX accumulation at DSB sites and feedback activation of ATM via NBS1 ubiquitination both in vitro and in vivo. Our results clearly implicate a Peli1-mediated ubiquitination process in the accumulation of DDR proteins and subsequent activation of HR repair (Fig. 10).

## Methods
**Cell cultures**. Primary MEFs were isolated from 13.5 days post coitum embryos of Peli1$^{+/+}$ (WT) and Peli1$^{\Delta E4/\Delta E4}$ (ΔE4) mice (Supplementary Information). MEFs were cultured for a maximum of five passages. Primary ATM WT (+/+) and KO (−/−) MEFs were kind gifts from Dr. Young Soo Lee (Ajou University School of Medicine, Suwon, South Korea). Primary MEF cells were grown in Dulbecco's modified Eagle's medium (DMEM; WelGENE, Gyeongsan, South Korea)

supplemented with 20% fetal bovine serum (FBS; Hyclone, Logan, UT, USA) and antibiotics at 37 °C in a humidified atmosphere containing 5% $CO_2$. HEK293T, U2OS, and HeLa were grown in DMEM medium supplemented with 10% FBS and antibiotics at 37 °C in a humidified atmosphere containing 5% $CO_2$. DR-GFP (pHPRT-DRGFP expressing stable cell lines based on U2OS cells), EJ-GFP (pimEJ5GFP expressing stable cell lines based on U2OS cells), and U2OS-DSB reporter (U2OS 2–6–3) cells were grown in DMEM medium supplemented with 10% FBS and antibiotics at 37 °C in a humidified atmosphere containing 5% $CO_2$. HeLa H2AX (WT and KO) cells were provided by Dr. Makoto Nakanishi (University of Tokyo, Japan).

**Antibodies and reagents**. A list of all antibodies used in this study can be found in Supplementary Table 1. The following reagents were used: ATM inhibitor KU59933 (Tocris), ATR inhibitor VE-831, DNA-PKs inhibitor KU57788 (NU7441), PARP inhibitor PJ34 (Selleckchem, Houston, TX, USA), SUMO inhibitor 2D-08 (Sigma, St. Louis, MO, USA), MG132, cycloheximide, and dimethyl sulfoxide (AG Scientific, San Diego, CA, USA).

**Plasmid construction, siRNAs, and transfection**. FL cDNA sequence of human Peli1 was PCR amplified using oligo-dT primers. Peli1 ΔFHA1 lacked the FHA1 domain. Peli1 ΔFHA2 lacked the FHA2 domain. Peli1 ΔC included 280 N-terminal amino acids but lacked the C-terminal RING domain. Peli1 FL, Peli1 ΔC, Peli1 C, and ΔE4 were subcloned into Myc- (pcDNA3-Myc), GFP- (pEGFP-C1), or RFP (pTurbo-RFP-C1)-tagged fusion plasmids. To generate Peli1 RING mutants and phospho-dead mutant (diagram shown in Figs. 3d and 6b), and NBS1 ubiquitin lysine mutants (diagram in Fig. 7b), site-directed mutagenesis was carried out using QuickChange site-directed mutagenesis kit (Stratagene, San Diego, CA, USA). All mutation sites were evaluated by DNA sequencing analysis. shRNAs or siRNAs targeting human Peli1 (shPeli1), 3′-UTR Peli1 (3′-UTR shPeli1), luciferase (shLuc, as a control) or 3′-UTR Peli1 (siRNA), 3′-UTR NBS1 (siNBS1), and UBC13

(siUBC13 #1 and #2) were synthesized using pSuper vector (Oligoengine, Seattle, WA, USA) according to gene-specific sequences. shRNA targeting sequences are listed below: 5′-GGGTTCAACACACTAGCAT-3′ is for shPeli1, 5′-GCTCCTTT GGATATGCAATTT-3′ is for 3′-UTR shPeli1, and 5′-GGAATATGTGAGCCTG GGCGCC-3′ for luciferase (shLuc). siRNA-targeting sequences are listed below: 5′-TTCTCCGAACGTGTCACGTTT-3′ is for negative control (siCTRL), 5′-GCTCC TTTGGATATGCAATTT-3′ is for Peli1-targeting 3′-UTR (siPeli1 3′-UTR), 5′-GGCTATATGCCATGAATAA-3′ and 5′-CCAGATGATCCATTAGCAA-3′ are for UBS13 targeting siUBC13 #1 and #2, and 5′-GCTTATTTAGAGTCCTAGTTT-3′ is for NBS1-targeting 3′-UTR (siNBS1 3′-UTR).

pcDNAs encoding GFP-tagged human, GFP-tagged Peli2 (GFP- Peli2), and GFP-tagged Peli3 (α and β; GFP-Peli3α and GFP-Peli3β) were PCR amplified using oligo-dT primers. Mammalian expression clones of Flag-tagged NBS1 WT, ΔFHA, ΔBRCT1, ΔBRCT2, ΔMA, and ΔC plasmids were obtained from Hui-Kuan Lin (Department of Cancer Biology, Wake Forest University School of Medicine, Winston-Salem, NC, USA). GFP-tagged NBS1 WT was kindly gifted by Xiaochun Yu (Division of Molecular Medicine and Genetics, Department of Internal Medicine, University of Michigan, Ann Arbor, MI, USA). Myc-tagged MRE11, Myc-tagged RAD50, Myc-tagged NBS1, HA-tagged ubiquitin (HA-Ub, the pEGFP-N3 vector encoding TAP (Strep-Flag-tagged)), His-tagged ubiquitin (His-Ub), and YFP-MRE11 were kindly gifted from Hong Tae Kim (Sungkyunkwan University, Suwon, South Korea). FLAG-H2AX (WT, S139A, and S139E) and H2AX-GFP (WT, S139A, and S139E) constructs were generated by Gateway LR coning system from pEntry H2AX vector. 5′-AAGGCCACCCAGGCCGCCCAGGAGTC-3′ is for pEntry H2AX S139A and 5′-AAGGCCACCCAGGCCGAACAGGAGTC is for S139E. The following plasmids were obtained from Addgene (http://www.addgene.org): pBAD-I-SceI (Addgene plasmid #60960), pimEJ5GFP (Addgene plasmid #44026), pHPRT-DRGFP (Addgene plasmid #26476), pLenti CMV/TO GFP-MDC1 (779–2) (Addgene plasmid #26285), pEGFP-C1-FLAG-Ku80 (Addgene plasmid #46958), pcDNA5-FRT/TO-eGFP-53BP1 (Addgene plasmid #60813), and pCW-GFP-CtIP (GFP-CtIP; Addgene plasmid #71109). For site-directed mutagenesis, all constructs were generated with Muta-Direct Site Directed Mutagenesis Kit (iNtRON Biotechnology, Seongnam, South Korea) using GFP-tagged NBS1 or GFP-tagged Peli1 as a template according to the manufacturer's standard procedures. For transient transfection, cells were electroporated using a microporator (Digital Biotechnology, Seoul, South Korea).

**Immunoblotting analysis and immunoprecipitation assays.** For immunoblotting analysis, cells were collected and lysed in nuclear extraction buffer [20 mM HEPES pH 7.6, 20% glycerol, 250 mM NaCl, 1.5 mM MgCl2, 0.1% Triton X-100, 1 mM phenylmethylsulfoyl fluoride (PMSF), 1 mM dithiothreitol (DTT), and protease inhibitor cocktail (Roche, Basel, Switzerland)]. Equal amounts of protein were separated by SDS-polyacrylamide gel electrophoresis and analyzed by immunoblotting with indicated antibodies. For S-tag pull-down assay, 293T cells were adapted to suspension conditions and lysed in NETN buffer (100 mM NaCl, 1 mM EDTA, 20 mM Tris-HCl pH 8.0, 0.5% NP-40) containing 1 mM PMSF, 1 mM DTT, and protease inhibitor cocktail. Supernatants were incubated with streptavidin-Sepharose beads (Amersham Biosciences, Piscataway, NJ, USA) for 8 h at 4 °C and bound proteins were analyzed via immunoblotting. For immunoprecipitation, 293T, HeLa, or U2OS cells were lysed in NETN buffer and incubated at 4 °C for 30 min. These lysates were cleared by centrifugation and incubated with anti-Myc, anti-GFP, anti-NBS1, anti-Peli1, anti-ATM antibodies, or normal IgG (control). They were then incubated with protein A/G agarose beads followed by centrifugation and immunoblotting. Full-size, uncropped scan images of immunoblots are available in Source Data file.

**In vivo ubiquitination assays.** 293T cells were transfected with GFP, GFP-tagged Peli1 FL, Peli1 ΔC or Peli1 C, Flag-tagged NBS1 WT or Flag-NBS1 mutants, and HA-tagged ubiquitin (HA-Ub) plasmid in combination. At 48 h post transfection, cells were collected and cell lysates were collected into two aliquots. One aliquot (10%) was used for conventional immunoblotting. The other aliquot (90%) was used for immunoprecipitation with anti-Flag or anti-Myc antibodies and analyzed by immunoblotting. For autoubiquitination of Peli1, GFP-fused Peli1 WT, H313A, C336A, S121A/T127A mutants, and HA-Ub plasmid were transfected into 293T cells. At 48 h post transfection, cells were treated with IR. GFP-Peli1 proteins immunoprecipitated with anti-GFP antibody were then detected with indicated antibodies. For NBS1 ubiquitination by Peli1, GFP-Peli1 constructs alone or with FLAG-NBS1 and UBC13 siRNAs (#1 and #2) were introduced into 293T cells or U2OS cells. Levels and lysine specificity of NBS1 ubiquitination were evaluated with indicated antibodies.

**Clonogenic cell-survival assay.** Peli1 WT and Peli1 ΔE4 MEF cells, and Peli1-depleted (shPeli1 or 3′-UTR shPeli1) U2OS cells were treated with various doses of IR and split into 12-well plates (2000 cells per well). At 14 days after IR treatment, viable cells were fixed and stained with crystal violet. The optical density was measured spectrophotometrically at 490 nm. Each experiment was repeated at least three times.

**Laser micro-irradiation, imaging, and immunofluorescence.** Single-strand breaks or DSBs of DNA were induced using a Nikon A1 laser microdissection system (Nikon). U2OS cells were transfected with indicated GFP-tagged expression plasmids and seeded onto glass-bottom dishes (SPL Life Sciences, Co., Pocheon, South Korea). At 36 h post transfection, cells were incubated with 10 μM 5-brome-2′-deoxyuridine for 24 h before laser-induced DSBs. Cells were subjected to laser-induced DSBs with two different settings [1 s/16 lines (Figs. 1h, 4g, and 7a, d) and 3 s/32 lines] using ×60 oil objective. Two different intensity settings for micro-irradiation showed almost the same results for all experiments. Fixed wavelength of ultraviolet laser (405 nm) was used for laser microdissection in a temperature-controlled chamber with CO2 supplied. Time-lapse images were captured and fluorescence intensities of micro-irradiated areas relative to non-irradiated areas within the cell nucleus were calculated using NIS elements C software (Nikon, Melville, NY, USA). Each data series was normalized against baseline values. For immunofluorescence studies, cells were seeded onto glass-bottom dishes (SPL Life Sciences, Co., Pocheon, South Korea) and laser micro-irradiated. Ten minutes or 30 min later, cells were fixed with 4% paraformaldehyde (PFA). Cells were permeabilized with 0.5% Triton-X for 5 min and blocked in 3% bovine serum albumin for 30 min. Cells were incubated with indicated primary antibodies and Alexa-conjugated secondary antibodies (Invitrogen, Carlsbad, CA, USA) for 2 h each at room temperature. Cells were then stained with 4′,6-diamidino-2-phenylindole (DAPI) before mounting onto slides. For immunofluorescence studies, cells were fixed in 4% PFA for 5 min. These fixed cells were washed with phosphate-buffered saline (PBS), permeabilized in PBS containing 0.1% Triton X-100, and incubated with appropriate primary antibodies at room temperature for 2 h. After washing, cells were further incubated with goat anti-mouse IgG conjugated with either fluorescein isothiocyanate or Cy5 at room temperature for 1 h. Cells were then washed, stained with Hoechst dye to visualize DNA, and viewed under a confocal microscope (Nikon A1, Japan; Zeiss 510 Meta, Germany).

**Chromatin fractionation.** 293T cells were transfected with NBS1 3′-UTR-targeting siRNA. At 24 h after transfection, cells were transfected again with GFP-NBS1 WT, K665/683R, or K686/690R mutants. After 48 h, DSB-mimetic drug phleomycin (50 μg/ml) was used to treat cells. At 30 min after washing, whole-cell lysates were extracted with NETN buffer (50 mM Tris-Cl pH 8.0, 0.5% nondinet P-40, 150 mM NaCl) including proteases and phosphatase inhibitor cocktail (Roche). Chromatin fractions were isolated with chromatin buffer [50 mM Tris-Cl pH 8.0, 0.5% Nondinet P-40, 150 mM NaCl, MNase (50 U), Benzonase (50 U)] at 37 °C for 20 min. Soluble fractions isolated from whole-cell lysates and chromatin fractions were used for western blotting assay.

**HR and NHEJ repair assays.** HEK293T cells were transfected with control (shLuc) or Peli1 shRNA (shPeli1) combination with pHPRT-DRGFP (HR reporter gene) or pimEJ5GFP (NHEJ reporter gene) plasmids. At 24 h post transfection, cells were then electroporated with 10 μg of pCBA-I-SceI plasmid using a microporator (Digital Biotechnology, Seoul, Korea). At 48 h post transfection, cells were collected and analyzed using flow cytometry. The frequency of GFP+ cells was determined from a portion of the sample by FACS Canto II cytometer. DR-GFP or EJ-GFP-stable cell lines were transfected with control (shLuc) or Peli1 shRNA (shPeli1). At 24 h post transfection, cells were then electroporated with 10 μg of pCBA-I-SceI plasmid using a microporator (Digital Biotechnology, Seoul, South Korea). At 48 h post transfection, cells were collected and analyzed using flow cytometry. The frequency of GFP+ cells was determined from a portion of the sample by FACS Canto II cytometer. For recovery of HR repair by reintroducing Myc-Peli1 WT, RING mutants, or phospho-dead mutant, U2OS DR-GFP-stable reporter cells were transfected with Peli1 3′-UTR-targeting siRNA. At 24 h post transfection, Myc-Peli1 WT, mutant constructs, and I-SceI endonuclease plasmids were transfected again into reporter cells. GFP-positive cells were counted by fluorescence-activated cell sorting analysis at 48 h post transfection. ATM inhibitor (KU-60019, TargetMol) was used as a positive control for HR repair efficiency.

**FokI assays.** mCherry-LacI-FokI was cotransfected with indicated GFP-tagged Peli1 plasmids and YFP-MRE11 (positive control) expression vector were transfected into U2OS-DSB reporter cells (U2OS 2–6–3). At 48 h post transfection, cells were fixed in 4% PFA for 5 min. These fixed cells were washed and permeabilized in PBS containing 0.1% Triton X-100. These cells were washed and stained with Hoechst dye to visualize DNA and confirm the recruitment of DSB-positive repair signals, γH2AX, 53BP1, RPA32, and RAD51 at DSB sites. To measure cell cycle dependency of GFP-Peli1 at DSB site, GFP-Peli1 was transfected into U2OS 2–6–3 cells. Cells were then fixed after 48 h. These fixed cells were immunostained with cyclinE or CyclinB1 antibodies and viewed under a confocal microscope.

**Metaphase chromosome spreading assays.** Peli1 WT and KO MEFs were treated with colcemid (100 ng/ml; Gibco, Carlsbad, CA, USA) for 6 h and mitotic cells were collected by shake-off. These cells were then incubated in a hypotonic buffer and fixed with Carnoy's solution. Cells in Carnoy's solution were dropped onto glass slides and dried at room temperature. Slides were stained with DAPI, mounted, and analyzed under a confocal microscope.

**Statistical analysis**. All values in this study are reported as mean ± SEM. Representative of an average of at least three independent experiments is shown. Student's $t$-test was used for statistical analyses. $P$-value < 0.05 was considered statistically significant. Significance is indicated by asterisk (*$P$ < 0.05, **$P$ < 0.01, ***$P$ < 0.001 vs. the control).

**Reporting Summary**. Further information on experimental design is available in the Nature Research Reporting Summary linked to this article.

## Data availability

All relevant data are available from the corresponding author on reasonable request. The uncropped western blottings and the source data underlying Figs. 1d, 1h, 7a, 7d, 8e, and 9e are provided as a Source Data file.

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

## Acknowledgements
We thank Drs. Hui-Kuan Lin (Wake Forest University School of Medicine), Xiaochun Yu (University of Michigan), Hong Tae Kim (Sungkyunkwan University), Young Soo Lee (Ajou University), and Ho Chul Kang (Ajou University) for supplying the materials. We also thank Joon-Sup Yoon and Kyung-Mo Kim for technical assistances. This study was supported by a National Research Foundation grant funded by the Korean government (MEST) (2017M2A2A7A01070267, 2017R1A2B3006776, and 2018R1D1A1B07048919).

## Author contributions
J.-H.J. designed and did the research, and prepared and wrote the part of manuscript. G.-H.H. designed and did the research. S.C., S.K., J.P., J.-K.L., Y.K., S.M., and J.-M.P. participated in data generation and analysis. T.-H.K. and H.L. provided materials and participated in data generation. T.-H.K. and H.C. designed the studies, analyzed data, and wrote the part of manuscript. C.-W.L. designed the studies, supervised the overall project, wrote the manuscript, and performed the final manuscript preparation. All authors provided feedback and agreed on the final manuscript.

## Additional information

**Competing interests:** The authors declare no competing interests.

