## [Peer Review File · Nature Communications]

Reviewers' comments:

Reviewer #1 (Remarks to the Author):

Ha et al Nat. Comm.

Ha et al describe a novel factor in the DNA damage response, Pellino 1 (Peli1), that is part of family of E3 ligases that mediate a variety of signaling events. Here the authors show that Peli1 localizes to sites of DNA double-strand breaks and that it is necessary for several events involving the ATM protein kinase and the MRN complex at these sites. Peli1 is shown to be required for efficient histone H2AX phosphorylation by ATM, ATM autophosphorylation, and for recruitment of the MRN complex through the Nbs1 component. Peli1 is phosphorylated, apparently by ATM, and this controls its DNA damage-induced autoubiquitination. Interestingly, the presence of Peli1 in cells controls the association of ATM with MRN as well as downstream events in DNA end processing, including homologous recombination. Overall, this is a comprehensive study that provides a large range of data supporting the role of this novel factor in the DNA damage response. This will be interesting to many people in the DNA damage field as well as potentially others who have interests in the Pellino family of proteins. I do have a few questions and comments about the data, however, and there are places where the manuscript contains overstated conclusions that should be modified.

Specific experimental points:

1. 8. p. 14: the conclusion that " these results indicate that Peli1 is required for ATM activation through NBS1 ubiquitination" is not justified here because the authors have not identified sites of Nbs1 ubiquitination and mutated them to determine the effects on ATM function. Also, it would need to be demonstrated that a point mutation eliminating E3 ligase activity in Peli1 also eliminates the effects on ATM, which is not done here. Either these experiments need to be included or the text needs to be modified (including the abstract and the title) to more accurately reflect what is demonstrated in this study.
2. Fig. 1c and other gH2AX blots in the paper: it would be good to show at least one blot with both H2AX (non phospho-specific) as well as the gH2AX (phospho-specific) signals.
3. Fig. 1d: quantitation of co-localizing foci is necessary here. Is the duration or intensity of the images shown for No IR different from that shown for 1 hr post-IR? Is Peli1 reproducibly observed at the nuclear periphery? It doesn't appear this way with the GFP-tagged protein in Fig. 1h.
4. With the mCherry-Foki/Peli1 colocalization in Fig. 1g, is there cell cycle specificity to this colocalization?
5. It appears that the deltaFHA1 and maybe also dFHA2 are largely excluded from the nucleus, based on the images shown in Fig. 2c and 2d. Also dFHA1 is expressed at a significantly lower level than the wt protein. If dFHA1 and 2 are expressed with a heterologous NLS, does this rescue their recruitment to damage sites?
6. Fig. S5d: The NLS of Nbs1 is reported to be in the region between BRCT2 and the Mre11-binding domain of Nbs1 (PMID 16188882). Are the deltaC or deltaMA Nbs1 mutants in the nucleus?
7. Fig. 5f, bottom: The Pel1 signal in the IP blot is just not convincing here because of the presence of the presumably nonspecific upper band.
8. It seems that the K63 ubiquitin is utilized much less efficiently than the wild-type ubiquitin. I am not convinced that K63 is entirely responsible for the functional effects of this protein. Have the

authors tried other lysines such as K6, K11, etc?

Issues related to the manuscript and figure labeling:

9. Fig 1a, b, manuscript p. 6: Need to have at least a minimal description of these cells, the phenotype, and what exactly is being quantified in the graph. Presumably the breaks measured are double-strand breaks, both chromatid and chromosome breaks, as observed in mitotic spreads?

10. p. 8, line 145: "reduced recruitment of γ H2AX to DSB foci ": γ H2AX is not recruited; the histone H2AX near the break site is phosphorylated.

11. Fig. 3e: the labeling should make it clear that wt GFP-Peli1 is expressed in the cells in lane 1; the difference is presumably only in the antibody used for the IP.

12. Fig. 3h: the labeling is also confusing here. All of the cells shown are expressing GFP-H2AX? Again the labeling should make it clear that wt GFP-Peli1 is expressed in the cells in lane 1, and the heading should say "GFP-H2AX", not just "GFP".

13. Fig. 6b: "K48 Ub" presumably refers to ubiquitin where K48 is the only available lysine, and similarly with K63. This should be made clear in the main text and in the legend.

14. There are grammatical issues throughout that should be corrected.

Reviewer #2 (Remarks to the Author):

In this manuscript Ha et al. report a role for the E3 ubiquitin ligase Pellino 1 (Peli1) in promoting accumulation of DNA repair factors at DNA double-strand break (DSB) sites to facilitate their repair by homologous recombination (HR). The authors show that Peli1 is recruited to DSBs in a manner involving its FHA domains and ATM-dependent phosphorylation, and that functional inactivation of Peli1 impairs robust activation of ATM-mediated signalling following DSB induction. The authors go on to show that this involves recognition of H2AX and NBS1 by the Peli1 FHA domains, and that Peli1 overexpression leads to increased NBS1 ubiquitination. Finally, they provide evidence that Peli1 regulates the stability of the MRE11-NBS1-RAD50 (MRN) complex, which amplifies ATM activation, to promote efficient HR and survival following DSB induction.

While the finding that Peli1 might function in the initial stages of ATM activation in the response to DSBs is novel and of potential importance, the manuscript in its current form falls short of convincingly demonstrating such an involvement. In particular, the presented dataset does not fully support the authors' conclusions and needs to be further substantiated on a number of points, as elaborated below, in order for this manuscript to be a strong candidate for publication in a journal like Nature Communications.

Major points:

1. A confounding issue when studying the Peli1 deltaE4 MEFs and mice is that this Peli1 mutant is clearly expressed at much lower levels than the wild-type protein. This makes it difficult to firmly conclude that the observed defects seen upon expression of this mutant is a specific consequence of the loss of the FHA domain and not Peli1 protein per se. In this respect, given that the Peli1 deltaE4 MEFs display markedly impaired ATM activation and DSB repair, it is surprising that the Peli1 deltaE4 mice appear to develop normally, unlike ATM- and H2AX-deficient mice. This raises

some doubts as to the relative physiological importance of Peli1 for ATM activation, also considering the statement that "Peli1 expression is highly suppressed under normal or non-pathological conditions" (line 377-378). Perhaps the authors could include data on the basic characterization of the Peli1 knockout and deltaE4 mice or, at the very least, discuss possible reasons for the lack of obvious phenotypes in these animals.

2. In a number of experiments, the intensity settings used for laser micro-irradiation appear to be abnormally high, as there is obvious bleaching of the GFP signal (e.g. the GFP- γ H2AX, GFP-NBS1 and GFP-MDC1 panels in Fig. 1h; all panels in Fig. 4g). Such extreme laser conditions could result in excessive DNA damage that non-specifically attracts chromatin-associated proteins. Indeed, total H2AX usually does not display net recruitment to DSB sites, unlike what is seen in Fig. 1h. The authors should repeat these experiments with more moderate laser intensity settings to ensure that a more physiological response to DSBs is induced. In several panels (e.g. Fig. 1h, 2c, 4a, 4e), the overall intensity of GFP-Peli1 visibly decreases over time, regardless of whether it is recruited to laser lines or not. Can the authors comment on the significance of this?

3. The notion that Peli1 interacts directly with H2AX in a manner that requires the FHA domain in Peli1 and S139 phosphorylation of H2AX is not convincing. First, it is well established that FHA domains interact specifically with phospho-threonine but not phospho-serine, which is inconsistent with Peli1 binding directly to H2AX pS139. Second, the observed "decreased" H2AX binding of the Peli1 deltaE4 mutant appears to be largely an effect of the lower expression level of this mutant (Fig. 4i).

4. In addition to the relatively crude approach of deleting entire protein domains or regions, the authors should study the impact of more subtle inactivating point mutations in the Peli1 FHA and RING domains on the ATM-mediated DSB response. This might potentially enable them to overcome the stability issues associated with expression of mutants lacking the FHA1 and FHA2 domains.

5. Mechanistically, how phosphorylation of S121 and T127 within the FHA2 domain facilitates Peli1 recruitment to DNA damage sites remains unclear. Do these modifications affect the ability of the FHA2 domain to bind H2AX and NBS1, or could it involve dimerization/oligomerization of Peli1? To conclude that S121/T127 phosphorylation of Peli1 enhances its E3 ligase activity (Fig. 3f), the authors would need to provide additional evidence that the ubiquitinated species present in GFP-Peli1 IPs represent auto-ubiquitination.

6. It is hard to be certain that the effects observed upon Peli1 depletion are specific, given the lack of rescue experiments. The study would be significantly stronger if the authors tested the ability of wild-type and mutant Peli1 alleles (FHA, RING and ATM phosphorylation-deficient mutants) to complement key phenotypes resulting from loss of endogenous Peli1 (e.g. IR sensitivity, ATM activation and DSB repair pathway efficiency).

7. The authors claim that Peli1 exerts its role in ATM activation by ubiquitinating NBS1 with K63 chains. The data supporting this potential mechanism are weak and need to be further substantiated. For instance, the authors should test the ability of FHA domain and S121/T127 phosphorylation-deficient mutants to promote NBS1 ubiquitination and seek to demonstrate Peli1-mediated NBS1 ubiquitination under conditions that do not involve overexpression of all factors. The evidence for Peli1 modifying NBS1 with K63-linked ubiquitin chains is entirely based on Fig. 6b, in which a K63-only ubiquitin mutant is used. While this mutant can be helpful for studying K63 ubiquitination, it is also notoriously unreliable with respect to recapitulating endogenous, physiological ubiquitination. Moreover, relative to wild-type ubiquitin, NBS1 ubiquitination is very inefficient in the presence of K63-only ubiquitin, suggesting that most of the NBS1 ubiquitin modifications induced by Peli1 overexpression are not K63-linked. To more convincingly demonstrate that Peli1 decorates NBS1 with K63 chains, the authors could perform ubiquitination experiments under conditions of Ubc13 depletion and probe for endogenous K63 ubiquitin chains using a K63-specific antibody. If Ubc13 knockdown indeed suppresses NBS1 ubiquitination upon Peli1 overexpression, the authors could test whether ATM activation is also compromised under these conditions.

8. Related to the above point: While Peli1 accumulates at DSBs in an ATM-dependent manner, a common trend throughout the paper is that Peli1 inactivation by shRNA-mediated depletion or expression of Peli1 deltaE4 leads to lower expression levels of a range of DSB-associated proteins

(incl. ATM, NBS1, MRE11, RAD50, and RAD51). This could indicate that rather than having a specific role in ubiquitinating NBS1, Peli1 may compromise ATM activity and HR via a more indirect involvement in maintaining the expression of the aforementioned proteins (unless of course the authors can provide evidence for a role of NBS1 ubiquitination in stabilizing these factors).

9. In Fig. 7, the authors used a single shRNA for Peli1 to show that its knockdown results in a decrease in homology-directed repair but not in NHEJ. Additional Peli1 shRNAs (for example the shRNA targeting the 3'-UTR used in Fig. 7i) should be used to rule out off-target effects. The authors also need to verify that Peli1 depletion does not impair HR efficiency simply by altering the cell cycle distribution.

10. In addition to Peli1, the authors show that other Peli proteins (Peli2, Peli3alpha and Peli3beta) also contain FHA domains and are recruited to DSB sites. Do these factors also contribute to ATM activation or is this function specific to Peli1?

Additional points:

11. It is unclear whether endogenous Peli1 is primarily a nuclear or cytoplasmic protein based on the provided images. For instance, in Fig. 1d, untreated cells display much more cytoplasmic Peli1 than cells subjected to IR. In Fig. 1e, endogenous Peli1 is exclusively nuclear, while in Fig. 1g Peli1 is again mostly cytoplasmic. It would be useful if the authors provided more insights into these localisation patterns (e.g. cell cycle- and DNA damage-dependencies), which might be relevant for the function of Peli1 in the context of ATM activation and the DNA damage response.

12. Several imaging-based experiments (e.g. Fig. 1d-g, Fig. 4d) need quantification.

13. The authors could provide more insight into the mechanism of Peli1 stabilization after DSBs (Fig. 1c).

14. Fig. 3e: Lane 3 in the IP is overloaded (higher GFP-Peli1 signal), which could explain the increased reactivity with the pSQ/TQ antibody.

15. Fig. 5c,d: The ability of ectopic Peli1 to rescue ATM signalling in Peli1 deltaE4 MEFs is not very convincing. In Fig. 5c, the higher p-ATM signal in Lane 4 could be due to more protein being loaded, as judged from the Actin blot.

16. Fig. 5f: Data shown in the lower panel is not convincing.

17. Line 109-110: The LacR-FokI nuclease does not induce a single DSB but instead generates numerous clustered DSBs in a single genomic locus. This should be corrected.

Reviewer #1

Specific experimental points:

1. 8. p. 14: the conclusion that "these results indicate that Peli1 is required for ATM activation through NBS1 ubiquitination" is not justified here because the authors have not identified sites of Nbs1 ubiquitination and mutated them to determine the effects on ATM function. Also, it would need to be demonstrated that a point mutation eliminating E3 ligase activity in Peli1 also eliminates the effects on ATM, which is not done here. Either these experiments need to be included or the text needs to be modified (including the abstract and the title) to more accurately reflect what is demonstrated in this study.

We would like to thank the reviewer for these valuable comments which raised a pertinent point and helped us improve our manuscript. To address this criticism, we performed a series of ubiquitination assay using three Peli1 mutants, H313A (HA) and C336A (CA) as ubiquitin ligase activity-dead mutants published very recently (Choi SW et al, Molecular Cell 2018 70: 920-935) and S121A/T127A (AA) as ATM-dependent phosphorylation-dead mutant. As shown in revised Figure 6, our ubiquitination assays revealed that levels of autoubiquitinated Peli1 in cells transfected with GFP-Peli1 WT were significantly increased by IR whereas those in cells transfected with GFP-Peli1 RING mutants (HA and CA) or phospho-dead mutant (AA) did not IR (revised Figures 6b-6c). These results support the notion that autoubiquitination of Peli1 might be dependent on its E3 ligase activity and ATM-mediated phosphorylation in response to IR.

In addition, we have provided new data sets demonstrating that Peli1 mediates K63-linked ubiquitination of NBS1 which is dependent on the ubiquitin ligase activity and ATM-dependent phosphorylation of Peli1. We have included new data in revised Figures 6d-6f and relevant descriptions on pages 14-15. Peli1 WT and point mutants (HA, CA and AA) were also used to determine if Peli1 activity could regulate phospho-ATM (p-ATM). Our results further confirmed that the E3 ligase activity of Peli1 is important for active retention of p-ATM at DSB sites (revised Supplementary Figures 8a-8b).

Notably, it has been reported that K665 and K683 residues are ubiquitinated by another E3 ligase, RNF8. Importantly, we found that Peli1-targeting conserved lysine residues (K686 and

K690) in NBS1 could be induced by Peli1-mediated ubiquitination through sequence alignment of amino acids. As shown in the revised Figure 7, Peli1-mediated NBS1 ubiquitination was clearly evident in NBS1 WT and K665/683R mutant, but not in NBS1 K686/690R mutant. Results of our microirradiation and chromatin retention experiments additionally revealed that NBS1 K686/690R mutant was unable to be retained at DSB sites whereas NBS1 WT and K665/683R mutant were efficiently retained (revised Figures 7d-7e). Together, these results support that Peli1 is required for DSB-induced ATM activation via NBS1 ubiquitination. Relevant statements are also included on pages 15-16 of the revised manuscript.

2. Fig. 1c and other gH2AX blots in the paper: it would be good to show at least one blot with both H2AX (non phospho-specific) as well as the gH2AX (phospho-specific) signals.

We agree with the reviewer. Therefore, we have added new blots of total H2AX (non phospho-specific) in revised Figures 1c, 2f, and 3f.

3. Fig. 1d: quantitation of co-localizing foci is necessary here. Is the duration or intensity of the images shown for No IR different from that shown for 1 hr post-IR? Is Peli1 reproducibly observed at the nuclear periphery? It doesn't appear this way with the GFP-tagged protein in Fig. 1h.

We have provided new data in revised Figure 1d using different Peli1-specific antibody. Peli1 mainly localizes in cytoplasm rather than in nucleus. Upon DNA damage, some portions of Peli1 protein translocated to the nucleus. Approximately 80% of them colocalized with DNA damage marker γ H2AX foci. The expression of Peli1 protein was detected by anti-Peli1 antibody (Santa Cruz Biotechnology, F-7, USA) which was able to detect both Peli1 and Peli2 proteins in the original manuscript. In the revised version of the manuscript, Peli1 was confirmed by using another anti-Peli1 specific antibody (Abcam, ab199336, USA).

4. With the mCherry-FokI/Peli1 colocalization in Fig. 1g, is there cell cycle specificity to this colocalization?

To address this important question, the revised manuscript now includes information of cell

cycle specificity of mCherry-FokI/Peli1 colocalization. Our results suggest that their colocalization might have occurred in G1 to S/G2 phase. We have performed additional cell cycle analyses and added relevant data into the revised Figure 1g and Supplementary Figure 3b. Relevant statements are added into page 8.

5. It appears that the deltaFHA1 and maybe also dFHA2 are largely excluded from the nucleus, based on the images shown in Fig. 2c and 2d. Also dFHA1 is expressed at a significantly lower level than the wt protein. If dFHA1 and 2 are expressed with a heterologous NLS, does this rescue their recruitment to damage sites?

We thank the reviewer for this interesting point. To address this comment, we have performed additional experiments by subcloning heterologous NLS to Peli1 Δ FHA1 and Peli1 Δ FHA2 to generate Peli1-NLS- Δ FHA1 and Peli1-NLS- Δ FHA2, respectively. However, these newly generated Peli1-NLS- Δ FHA1 and Peli1-NLS- Δ FHA2 mutants were still unable to translocate at laser stripes or at FokI site (revised Figures 2c and 2d and supplementary Figure 4), indicating that Peli1 FHA domain is essential for targeting Peli1 to DSB sites.

6. Fig. S5d: The NLS of Nbs1 is reported to be in the region between BRCT2 and the Mre11-binding domain of Nbs1 (PMID 16188882). Are the deltaC or deltaMA Nbs1 mutants in the nucleus?

To answer this important question, we have performed immunostaining again using FLAG-tagged NBS1 WT and truncated mutants. We have added new data into supplementary Figure 7e. The relevant statements are described on page 13. NBS1 Δ MA mutant was localized in the nucleus because this mutant still maintained the NLS. However, NBS1 Δ C mutant, for which the NLS signal was removed, was still localized in the nucleus partially.

7. Fig. 5f, bottom: The Peli1 signal in the IP blot is just not convincing here because of the presence of the presumably nonspecific upper band.

We apologize for this confusion. To address this comment, we have performed additional experiment using Trueblot secondary antibody which cannot detect non-specific bands (e.g.

heavy chains or light chains). We have replaced the old Fig. 5f with the new data in the revised Figure 5f.

8. It seems that the K63 ubiquitin is utilized much less efficiently than the wild-type ubiquitin. I am not convinced that K63 is entirely responsible for the functional effects of this protein. Have the authors tried other lysines such as K6, K11, etc?

We agree with the reviewer that we need to address this important question. Therefore, we have added new data to the revised Figure 6. Our new data strongly suggest that ubiquitination of NBS1 is significantly reduced by its RING mutants and ATM-dependent phosphorylation mutant in comparison with Peli1 WT. In addition, our results further show that Peli1-mediated NBS1 ubiquitination is K63-linkage-specific (revised Figure 6d). K63-linked NBS1 ubiquitination was further demonstrated by additional experiment through deposition of K63-specific E2 enzyme UBC13 (revised Figures 6e and 6f). Relevant statements are added into pages 14-15 of the revised manuscript.

Issues related to the manuscript and figure labeling:

9. Fig 1a, b, manuscript p. 6: Need to have at least a minimal description of these cells, the phenotype, and what exactly is being quantified in the graph. Presumably the breaks measured are double-strand breaks, both chromatid and chromosome breaks, as observed in mitotic spreads?

We have included the description of cells and mouse phenotypes in the revised manuscript (page 7) and revised supplementary Figures 5c and 5d. We have also corrected Figure 1b.

10. p. 8, line 145: "reduced recruitment of γ H2AX to DSB foci ": γ H2AX is not recruited; the histone H2AX near the break site is phosphorylated.

To avoid confusion, we have revised Figure 2g with new data from additional experiments. We have also modified the text accordingly (page 9).

11. Fig. 3e: the labeling should make it clear that wt GFP-Peli1 is expressed in the cells in lane 1; the difference is presumably only in the antibody used for the IP.

We agree with the reviewer. Therefore, we have revised Figure 3e with new data from additional experiment.

12. Fig. 6b: "K48 Ub" presumably refers to ubiquitin where K48 is the only available lysine, and similarly with K63. This should be made clear in the main text and in the legend.

We agree with the reviewer. Therefore, we have provided new data sets demonstrating that K63-linked ubiquitination of NBS1 could be mediated by Peli1. Our new data strongly support that the ubiquitination of NBS1 is significantly reduced by Peli1 RING mutants and phospho-dead mutant compared to Peli1 WT. In addition, our results further show that Peli1-mediated NBS1 ubiquitination is K63-linkage specific (revised Figure 6e). K63-linked NBS1 ubiquitination was further demonstrated by additional experiment through deposition of K63-specific E2 enzyme UBC13 (revised Figure 6e and 6f). Therefore, we have rephrased this statement on pages 14-15 of the revised manuscript as suggested by the reviewer.

14. There are grammatical issues throughout that should be corrected.

We thank the reviewer for pointing this out and we agree with the reviewer. Therefore, we have asked a professional English Editing Service (Harrisco) to fix our grammatical issues. 'Certificate of Editing' can be provided if necessary.

Reviewer #2

Major points:

1. A confounding issue when studying the Peli1 deltaE4 MEFs and mice is that this Peli1 mutant is clearly expressed at much lower levels than the wild-type protein. This makes it difficult to firmly conclude that the observed defects seen upon expression of this mutant is a specific consequence of the loss of the FHA domain and not Peli1 protein per se. In this respect, given that the Peli1 deltaE4 MEFs display markedly impaired ATM activation and DSB repair, it is surprising that the Peli1 deltaE4 mice appear to develop normally, unlike ATM- and H2AX-deficient mice. This raises some doubts as to the relative physiological importance of Peli1 for ATM activation, also considering the statement that "Peli1 expression is highly suppressed under normal or non-pathological conditions" (line 377-378). Perhaps the authors could include data on the basic characterization of the Peli1 knockout and deltaE4 mice or, at the very least, discuss possible reasons for the lack of obvious phenotypes in these animals.

We would like to thank the reviewer for these valuable comments which raised a pertinent point and helped us improve our manuscript. To address this comment, we have included information of Peli1 Δ E4 mice in the revised supplementary Figures 5c and 5d. Relevant statements are now included on page 9 of the revised manuscript.

2. In a number of experiments, the intensity settings used for laser micro-irradiation appear to be abnormally high, as there is obvious bleaching of the GFP signal (e.g. the GFP- γ H2AX, GFP-NBS1 and GFP-MDC1 panels in Fig. 1h; all panels in Fig. 4g). Such extreme laser conditions could result in excessive DNA damage that non-specifically attracts chromatin-associated proteins. Indeed, total H2AX usually does not display net recruitment to DSB sites, unlike what is seen in Fig. 1h. The authors should repeat these experiments with more moderate laser intensity settings to ensure that a more physiological response to DSBs is induced. In several panels (e.g. Fig. 1h, 2c, 4a, 4e), the overall intensity of GFP-Peli1 visibly decreases over time, regardless of whether it is recruited to laser lines or not. Can the authors comment on the significance of this?

We thank the reviewer for pointing this out. In response to this comment, the intensity settings used for laser micro-irradiation in our new experiments have been changed from 3 Sec and 32 lines to 1 Sec and 16 lines for GFP signals (revised Figures 1h, 4g, 7a, and 7d; revised supplementary Figures 6a and 8a). We used confocal microscope (Nikon A1) to detect live GFP image (no interval) after laser micro-irradiation which can lead to decrement of GFP signal under this condition. We were able to obtain almost the same result when reduced intensity settings were used for laser micro-irradiation. We have provided relevant statements on page 29 (Materials and Methods section) of the revised manuscript.

3. The notion that Peli1 interacts directly with H2AX in a manner that requires the FHA domain in Peli1 and S139 phosphorylation of H2AX is not convincing. First, it is well established that FHA domains interact specifically with phospho-threonine but not phospho-serine, which is inconsistent with Peli1 binding directly to H2AX pS139. Second, the observed "decreased" H2AX binding of the Peli1 deltaE4 mutant appears to be largely an effect of the lower expression level of this mutant (Fig. 4i).

As commented by this reviewer, FHA domain has been well established which directly interacts with phospho-threonine residue. However, Kobayashi J. et al (Current Biology 2002 12:1846-1851) have also shown direct interaction between phospho-serine residue and FHA domain in addition to phospho-threonine specificity with FHA interaction. We also found that phospho-serine of H2AX could directly interact with FHA of Peli1 (revised Figure 4h).

To address this important comment, we have performed additional experiments. In condition showing similar expressions of GFP-Peli1 WT and $\Delta E4$ mutant in 293T cells, we observed interaction of Peli1 WT, but not Peli1 $\Delta E4$ mutant, with FLAG-H2AX, indicating that Peli1-H2AX interaction might be dependent on the FHA domain (revised Figure 4i).

4. In addition to the relatively crude approach of deleting entire protein domains or regions, the authors should study the impact of more subtle inactivating point mutations in the Peli1 FHA and RING domains on the ATM-mediated DSB response. This might potentially enable them to overcome the stability issues associated with expression of mutants lacking the FHA1 and FHA2 domains.

We thank the reviewer for pointing this out and we agree with the reviewer. Therefore, we have performed a series of ubiquitination assays using three Peli1 mutants: H313A (HA) and C336A (CA) as ubiquitin ligase activity-dead mutants published very recently (Choi SW et al, Molecular Cell 2018 70: 920-935) and S121A/T127A (AA) as ATM-dependent phosphorylation-dead mutant. As shown in revised Figure 6, our ubiquitination assays revealed that levels of autoubiquitinated Peli1 in cells transfected with GFP-Peli1 WT were significantly increased by IR whereas those in cells transfected with GFP-Peli1 RING mutants (HA and CA) or phospho-dead mutant (AA) did not (revised Figures 6b-6c). These results support notion that autoubiquitination of Peli1 might be dependent on its E3 ligase activity and ATM-mediated phosphorylation in response to IR.

We have also provided new data sets demonstrating that Peli1 mediates K63-linked ubiquitination of NBS1 which is dependent on the ubiquitin ligase activity and ATM-dependent phosphorylation in Peli1. We have included new data in revised Figures 6d-6f and relevant descriptions on pages 14-15. Peli1 WT and point mutants (HA, CA and AA) were also used to determine if Peli1 activity could regulate phospho-ATM (p-ATM). We have further confirmed that Peli1 E3 ligase activity is important for the active retention of p-ATM at DSB sites (revised supplementary Figures 8a and 8b).

Notably, it has been reported that K665 and K683 residues are ubiquitinated by RNF8, another existing E3 ubiquitin ligase. Importantly, we found another conserved lysine residue in NBS1 that could be induced by Peli1-mediated ubiquitination through sequence alignment of amino acids. As shown in the revised Figure 7, Peli1-mediated NBS1 ubiquitination was clearly evident in NBS1 WT and K665/683R mutant, but not in NBS1 K686/690R mutant. Results of the microirradiation and chromatin retention experiments additionally revealed that NBS1 K686/690R mutant was unable to be retained at DSB sites whereas NBS1 WT and K665/683R mutant were efficiently retained (revised Figures 7d and 7e). Taken together, these results support that Peli1 is required for DSB-induced ATM activation via NBS1 ubiquitination. Relevant statements are also included on pages 15-16 of the revised manuscript.

5. Mechanistically, how phosphorylation of S121 and T127 within the FHA2 domain facilitates Peli1 recruitment to DNA damage sites remains unclear. Do these modifications

affect the ability of the FHA2 domain to bind H2AX and NBS1, or could it involve dimerization/oligomerization of Peli1? To conclude that S121/T127 phosphorylation of Peli1 enhances its E3 ligase activity (Fig. 3f), the authors would need to provide additional evidence that the ubiquitinated species present in GFP-Peli1 IPs represent auto-ubiquitination.

In response to this important comment, we have performed additional experiments which revealed that autoubiquitination of Peli1 was augmented under DNA damage conditions (IR). However, DSB-induced Peli1 autoubiquitination was attenuated by the mutation of ATM-dependent phosphorylation site in Peli1 (S121A/T127A) (revised Figures 6b and 6c). In addition, unlike Peli1 WT, Peli1 S121A/T127A mutant was unable to induce K63-linked NBS1 ubiquitination. We further confirmed that Peli1 E3 ligase activity was important for the active retention of p-ATM at DSB site (revised supplementary Figures 8a and 8b). Taken together, these results support the notion that Peli1 recruitment to DSB site is triggered by ATM-mediated phosphorylation.

We found that endogenous Peli1 or GFP-Peli1 and RFP-Peli1 failed to move to laser strips in H2AX KO cells (revised Figures 4d and 4g). We also found that Peli1 Δ E4 and H2AX-Ser139 dead-mutant could not interact with FLAG-H2AX or RFP-Peli1, respectively (revised Figures 4h and 4i). In addition, our new data indicate that NBS1 ubiquitination could be regulated by ATM-dependent Peli1 phosphorylation. We have added our new data to revised Figures 7d and 7e.

6. It is hard to be certain that the effects observed upon Peli1 depletion are specific, given the lack of rescue experiments. The study would be significantly stronger if the authors tested the ability of wild-type and mutant Peli1 alleles (FHA, RING and ATM phosphorylation-deficient mutants) to complement key phenotypes resulting from loss of endogenous Peli1 (e.g. IR sensitivity, ATM activation and DSB repair pathway efficiency).

In response to this important comment, we have performed a series of ubiquitination assay using three Peli1 mutants, RING mutants [H313A (HA) and C336A (CA)] and phospho-dead mutant [S121A/T127A (AA)]. We have added new data to revised Figures 6b-6f and supplementary Figures 8a and 8b.

In addition to Peli1 rescue experiment in original Figures 5c and 5d (which are replaced with new data in the revised Figures 5c and 5d), our new rescue experiments using Peli1 WT, HA, CA, and AA mutants in Peli1-depleted cells showed that only Peli1 WT, but not Peli1 HA, CA, or AA mutants, was able to rescue p-ATM signal at DSB sites (revised supplementary Figures 6a and 6b). Relevant statements are also included on page 16 of the revised manuscript.

We further compared the effect of ectopic expression of Peli1 WT, HA, CA, and AA mutants on HR repair. As expected, under condition that the expression of Peli1 WT significantly recovered HR repair in endogenous Peli1-depleted cells, Peli1 HA, CA, or AA mutant was unable to rescue HR repair, indicating that Peli1 contributed to HR-mediated repair of DSB. We have presented new data in revised Figure 9e. Relevant statements are included on page 18 of the revised manuscript.

7. The authors claim that Peli1 exerts its role in ATM activation by ubiquitinating NBS1 with K63 chains. The data supporting this potential mechanism are weak and need to be further substantiated. For instance, the authors should test the ability of FHA domain and S121/T127 phosphorylation-deficient mutants to promote NBS1 ubiquitination and seek to demonstrate Peli1-mediated NBS1 ubiquitination under conditions that do not involve overexpression of all factors. The evidence for Peli1 modifying NBS1 with K63-linked ubiquitin chains is entirely based on Fig. 6b, in which a K63-only ubiquitin mutant is used. While this mutant can be helpful for studying K63 ubiquitination, it is also notoriously unreliable with respect to recapitulating endogenous, physiological ubiquitination. Moreover, relative to wild-type ubiquitin, NBS1 ubiquitination is very inefficient in the presence of K63-only ubiquitin, suggesting that most of the NBS1 ubiquitin modifications induced by Peli1 overexpression are not K63-linked. To more convincingly demonstrate that Peli1 decorates NBS1 with K63 chains, the authors could perform ubiquitination experiments under conditions of Ubc13 depletion and probe for endogenous K63 ubiquitin chains using a K63-specific antibody. If Ubc13 knockdown indeed suppresses NBS1 ubiquitination upon Peli1 overexpression, the authors could test whether ATM activation is also compromised under these conditions.

In response to these important comments, we have performed additional experiments provided additional results demonstrating that Peli1 mediates K63-linked ubiquitination of NBS1

which is dependent on the ubiquitin ligase activity and ATM-dependent phosphorylation in Peli1. We have included new data in revised Figures 6d-6f. Relevant descriptions are shown on pages 14-15. Peli1 WT and RING mutants (HA and CA) were also used to determine if Peli1 activity could regulate p-ATM. We found that Peli1 E3 ligase activity was important for the active retention of p-ATM at DSB sites (revised supplementary Figures 8a and 8b).

Notably, it has been reported that K665 and K683 residues are ubiquitinated by another E3 ligase RNF8. Importantly, we found another conserved lysine residue in NBS1 which could be induced by Peli1-mediated ubiquitination through sequence alignment of amino acids. As shown in revised Figure 7, Peli1-mediated NBS1 ubiquitination was clearly evident in NBS1 WT and K665/683R mutant, but not in NBS1 K686/690R mutant. Our microirradiation and chromatin retention experiments additionally showed that NBS1 K686/690R mutant was unable to be retained at DSB sites whereas NBS1 WT and K665/683R mutant were efficiently retained (revised Figures 7d and 7e). Together, these results support that Peli1 is required for DSB-induced ATM activation via NBS1 ubiquitination. Relevant statements are also included on pages 15-16 of the revised manuscript.

8. Related to the above point: While Peli1 accumulates at DSBs in an ATM-dependent manner, a common trend throughout the paper is that Peli1 inactivation by shRNA-mediated depletion or expression of Peli1 deltaE4 leads to lower expression levels of a range of DSB-associated proteins (incl. ATM, NBS1, MRE11, RAD50, and RAD51). This could indicate that rather than having a specific role in ubiquitinating NBS1, Peli1 may compromise ATM activity and HR via a more indirect involvement in maintaining the expression of the aforementioned proteins (unless of course the authors can provide evidence for a role of NBS1 ubiquitination in stabilizing these factors).

We thank the reviewer for pointing this out. To address this comment, we have extensively compared expression levels of DSB-associated proteins including ATM, NBS1, MRE11, RAD50, and RAD51 between Peli1 WT and $\Delta E4$ MEFs and presented revised data in Figures 8b and 9a. In addition, we have included new data (revised Figures 6 and 7, and supplementary Figures 6 and 8). In particular, our new rescue experiments using Peli1 WT and HA, CA, and AA point mutants in endogenous Peli1-depleted cells showed that only

Peli1 WT, but not Peli1 HA, CA, or AA point mutant, was able to rescue p-ATM signal at DSB sites (revised Figures 6d and revised supplementary Figure 6a and 6b). In addition, the mutation of highly conserved K686 and K690 residues in NBS1 (NBS1 K686/690R mutant) lost Peli1-mediated NBS1 ubiquitination (revised Figures 7b and 7c). We additionally found that NBS1 K686/690R mutant was unable to be recruited to DSB sites whereas NBS1 WT and K665/683R mutant were efficiently recruited (revised Figures 7d and 7e). With our additional data, these results support the notion that Peli1 is required for DSB-induced ATM activation via NBS1 ubiquitination.

9. In Fig. 7, the authors used a single shRNA for Peli1 to show that its knockdown results in a decrease in homology-directed repair but not in NHEJ. Additional Peli1 shRNAs (for example the shRNA targeting the 3'-UTR used in Fig. 7i) should be used to rule out off-target effects. The authors also need to verify that Peli1 depletion does not impair HR efficiency simply by altering the cell cycle distribution.

To check whether depletion of Peli1 might impair HR efficiency possibly by altering cell cycle distribution, we first examined cell cycle profile of Peli1-depleted cells generated by two different methods: Peli1-targeting shRNA or siRNA (3'-UTR). As shown in revised supplementary Figures 12g and 12h, Peli1 depletion did not alter cell cycle progression. We have included relevant statement on page 18 of the revised manuscript.

In addition, we alternatively employed Peli1 point mutants, RING mutants [H313A (HA) and C336A (CA)], and phospho-dead mutant [S121A/T127A (AA)]. Using these mutants, our rescue experiments showed that Peli1 HA, CA, and AA mutants failed to rescue p-ATM signal at DSB sites (revised supplementary Figures 8a and 8b). Furthermore, in condition that ectopic expression of Peli1 WT significantly recovered HR repair in endogenous Peli1-depleted cells, Peli1 HA, CA, or AA mutant was unable to rescue HR repair (revised Figure 9e and revised supplementary Figure 12i).

10. In addition to Peli1, the authors show that other Peli proteins (Peli2, Peli3alpha and Peli3beta) also contain FHA domains and are recruited to DSB sites. Do these factors also contribute to ATM activation or is this function specific to Peli1?

This is a very interesting comment. We are in the middle of performing experiments by generating various knock-out cell lines of Pellino family proteins. However, we apologize that we are unable to address this issue due to time constraints.

Additional points:

11. It is unclear whether endogenous Peli1 is primarily a nuclear or cytoplasmic protein based on the provided images. For instance, in Fig. 1d, untreated cells display much more cytoplasmic Peli1 than cells subjected to IR. In Fig. 1e, endogenous Peli1 is exclusively nuclear; while in Fig. 1g Peli1 is again mostly cytoplasmic. It would be useful if the authors provided more insights into these localisation patterns (e.g. cell cycle- and DNA damage-dependencies), which might be relevant for the function of Peli1 in the context of ATM activation and the DNA damage response.

We agree with the reviewer. Therefore, we have performed additional experiments and provided new data in revised Figures 1d-1g and supplementary Figures 3a and 3b. Relevant statements are also included in the revised manuscript.

In our study, the expression of Peli1 protein was observed by using anti-Peli1 antibody (Santa Cruz Biotechnology, F-7, USA) which was able to detect both Peli1 and Peli2 proteins in the original manuscript. However, in the revised version of the manuscript, Peli1 was confirmed by using another anti-Peli1 specific antibody (Abcam, ab199336, USA).

12. Several imaging-based experiments (e.g. Fig. 1d-g, Fig. 4d) need quantification.

We thank the reviewer for pointing this out and we agree with the reviewer. Therefore, we have included quantification for revised Figures 1d-1g and revised Figure 4d.

13. The authors could provide more insight into the mechanism of Peli1 stabilization after DSBs (Fig. 1c).

Recent studies have unveiled a critical role of Peli1 in activating toll-like receptor (TLR) and/or T-cell receptor (TCR) signaling-mediated pro-inflammatory gene expression

(Moynagh PN, Nature Rev Immunology 2014 154: 122-131; Chang M et al, Nature Immunology 2009 10: 1089-1095; Jin W et al, Cell Mol Immunology 2012 9: 113-122). Notably, Peli1 expression is highly suppressed under normal or non-pathological situations. In contrast, pathogenic conditions promote Peli1 expression. For instance, Peli1 expression is upregulated in patients with neutrophilic asthma (Baines KJ et al, J Allergy Clin Immunol 2011 127: 153-160) and those with diffuse large B cell lymphoma (Park HY et al, J Clinical Investigation 2014 124: 4976-4988). Interestingly, Peli1 expression is activated in response to various receptor-mediated signaling events such as those associated with TLR, TCR, and BCR. In this manuscript, for the first time, it is unveiled that Peli1 is activated by DNA DSB signal.

In response to this important comment, we additionally provided evidences that Peli1 was activated by ATM-mediated phosphorylation and recruited to DSB sites in ATM- and γ H2AX-dependent manners. Particularly, our new data revealed that autoubiquitination of Peli1 was augmented in DNA damage conditions (IR). However, DSB-induced Peli1 autoubiquitination was attenuated by mutation of ATM-dependent phosphorylation in Peli1 (S121A/T127A) (revised Figures 6b and 6c). In addition, unlike Peli1 WT, Peli1 S121A/T127A mutant was unable to induce K63-linked NBS1 ubiquitination and retain p-ATM at DSB site (revised supplementary Figures 8a and 8b). Taken together, these results support the notion that Peli1 recruitment to DSB site is triggered by ATM-mediated phosphorylation.

14. Fig. 3e: Lane 3 in the IP is overloaded (higher GFP-Peli1 signal), which could explain the increased reactivity with the pSQ/TQ antibody.

We thank the reviewer for pointing this out. In response to this comment, we have replaced our results through a new experiment (revised Figure 3e).

15. Fig. 5c,d: The ability of ectopic Peli1 to rescue ATM signalling in Peli1 deltaE4 MEFs is not very convincing. In Fig. 5c, the higher p-ATM signal in Lane 4 could be due to more protein being loaded, as judged from the Actin blot.

We have replaced our results with a new experiment (revised Figure 5c).

16. Fig. 5f: Data shown in the lower panel is not convincing.

We have performed additional experiment using Trueblot secondary antibody which is unable to detect non-specific bands (e.g. heavy chains or light chains). We have replaced our results with new data in revised Figure 5f.

17. Line 109-110: The LacR-FokI nuclease does not induce a single DSB but instead generates numerous clustered DSBs in a single genomic locus. This should be corrected.

We agree with the reviewer. Therefore, we have corrected our mistakes.

Reviewer #3

In all, the paper makes a fair case that Peli1 is recruited to DNA damage sites. However, the functional consequences of this recruitment are not explored or analyzed satisfactorily. It is not clear whether Peli1 has a bona fide role in repair. The m/s is top-heavy with laser stripes (which delivers an intense, probably lethal level of damage to the target cell) and other localization phenomena, mainly obtained via forced overexpression of the protein in cancer cell lines. However, the m/s fails to place sufficient emphasis on the quantitative functional significance of Peli1. Here, the authors have an excellent reagent—the Peli1 $\Delta 4$ homozygous mouse—the phenotype of which does not appear to support the major conclusions of this m/s. The authors may have failed to explore the Peli1 $\Delta 4$ homozygous mouse in sufficient depth.

We would like to thank the reviewer for these supportive comments and useful suggestions. As suggested by the reviewer, we have included the basic information of Peli1 $\Delta E4$ mice (revised supplementary Figures 5c and 5d) and MEFs (revised Figures 9f and 9g). Relevant statements are now included on pages 9 and 18 of the revised manuscript.

To avoid the possibility of abnormally high intensity settings used for laser micro-irradiation which may causes artificial distribution of DSB-associated proteins, we have provided new data by changing the intensity setting from 3 Sec and 32 lines to 1 Sec and 16 lines for GFP signals (revised Figures 1h, 4g, 7a, and 7d). However, almost the same results were obtained from such reduced intensity settings used for laser micro-irradiation. We have provided relevant statements on page 29 (Materials and Methods section) of the revised manuscript.

Specific comments:

1. The introduction discussing the mammalian DNA damage response lacks precision and detail. It would be helpful to recruit the input of an expert in mammalian DSB repair to review the introduction. Some of the written style is awkward and could be improved.

We have added more information about mammalian DNA damage response to the ‘Introduction’ section (page 4) of the revised manuscript.

2. The claim of a direct interaction with gamma-H2AX is not supported experimentally.

We thank the reviewer for pointing this out. We found that endogenous Peli1 or GFP-Peli1 and RFP-Peli1 failed to move to laser strips in H2AX KO cells (revised Figures 4d and 4g). We also found that Peli1 $\Delta E4$ and H2AX-Ser139 dead-mutant could not be interact with FLAG-H2AX and RFP-Peli1, respectively (revised Figures 4h and 4i). We have corrected the text accordingly (pages 11-12).

3. Why do many of the ectopically expressed fusion proteins studied localize to the nucleolus (e.g., Figs 1h, 4g)? What function does the cytoplasmic fraction of Peli1 perform? In general, there is a concern that a large fraction of the data is derived from analysis of overexpressed proteins, with the potential for introducing artifact.

To address this important comment, we have performed additional experiments and provided new data in revised Figures 1d-1g, 4g, and supplementary Figures 3a and 3b. Relevant statements are also included in the revised manuscript. In addition, to avoid the possibility of artificial distribution of DSB-associated proteins due to abnormally high intensity setting used for laser micro-irradiation, we employed mild intensity settings of 1 sec/16 lines instead of 3 sec/32 lines for detecting GFP signals (revised Figures 1h, 4g, 7a, and 7d).

In the original manuscript, the expression of endogenous Peli1 protein was observed by anti-Peli1 antibody (Santa Cruz Biotechnology, F-7, USA) which was able to detect both Peli1 and Peli2 proteins. However, in the revised version of the manuscript, Peli1 was confirmed by another Peli1 specific antibody (Abcam, ab199336, USA).

4. Fig 1. The data on chromosome breakage is inadequate. At a minimum, chromatid-type and chromosome-type errors must be scored. The bar chart (Fig 1b) is incongruent with the images shown, since it appears to claim 5 breaks per wt cell. The images show a chromosome-type error, rather than the chromatid-type errors that might be expected in an HR-defective cell. The FokI nuclear focus is not validated by an independent marker to identify the FokI target site.

We thank the reviewer for pointing this out and we agree with the reviewer. Therefore, we have modified Figure 1b. Regarding the second part of comment, notably mCherry-LacI-FokI recognized LacO (256x repeats) integrated into U2OS 2-6-3 chromosome 1p3.6. Therefore, mCherry-LacI-FokI directly interacted with LacO and then activated FokI by interacting with LacI-LacO, making a single DNA double-strand break in the locus. This system has been well established by Roger Greenberg's group (Shanbhag et al., Cell 2010 141: 970-981). We have also published a paper using this system (Min et al., Cell Cycle 2014 13: 666-677). To validate and identify FokI target site which caused a single double-strand break (sDSB) in this system, we checked factors involved in DNA damage and repair signaling. γ H2AX and 53BP1 were used as DNA damage signaling and NHEJ repair factor while RPA32 and RAD51 were used as HR factor. These factors were colocalized to sDSB. Since overexpression of GFP-Peli1 was colocalized with mCherry-LacI-FokI at sDSB (revised Figures 1f and 1g), an endogenous Peli1 also colocalized with mCherry-LacI-FokI and γ H2AX at sDSB (revised supplementary Figures 3a and 3b).

5. Fig 2. The reduced abundance of the Peli1 Δ FHA proteins makes comparison of their localization properties difficult to assess. An alternative interpretation of the data is that the Δ FHA proteins are not present at high enough levels to be detectable at damage sites.

We thank the reviewer for pointing this out. Although we do not exclude the possibility that Peli1 Δ E4 are somehow less stable than Peli1 WT, a number of results strongly support that the loss of FHA domain leads to the inability of Peli1 function as a DSB E3 ubiquitin ligase. Particularly, under conditions showing similar expressions of GFP-Peli1 WT and Δ E4 mutant into 293T cells, we observed that the interaction between Peli1 and FLAG-H2AX seemed to be dependent on its FHA domain (revised Figure 4i).

In addition, our new data showed that Peli1 Δ E4 MEFs displayed marked reductions of γ H2AX level (revised Figure 2f) with reduced and diffused recruitments of γ H2AX and phospho-ATM to DSB foci following exposure to irradiation (revised Figure 2g). To further evaluate biological functions of Peli1 in DNA damage repair, we performed a clonogenic survival assay in two different types of Peli1-depleted cells following IR. As expected, Peli1 Δ E4 MEFs displayed much higher sensitivity to IR than Peli1 WT MEFs (revised Figure 9g).

These results are very similar to Peli1-depleted cells by shPeli1 transfection (revised Figures 9h), supporting our conclusion that the recruitment of Peli1 to DSB sites is mediated by its FHA domain.

6. Fig 6 ubiquitination assays. The Ub-K63 signal appears less intense than the wt Ub signal. Why is this?

In response to this important comment, we have provided a number of additional results showing that Peli1 mediates K63-linked ubiquitination of NBS1 which is dependent on ubiquitin ligase activity and ATM-dependent phosphorylation of Peli1. We have included new data in revised Figures 6d-6f and relevant descriptions on pages 14-15. In addition, Peli1 WT and E3 ligase activity-dead mutants (HA and CA) were used to determine if Peli1 activity could regulate p-ATM. Our results revealed that Peli1 E3 ligase activity was important for the active retention of p-ATM at DSB sites (revised supplementary Figures 8a and 8b).

Notably, K665 and K683 residues have reported to be ubiquitinated by another existing E3 ubiquitin ligase RNF8. Importantly, we found that another conserved lysine residue in NBS1 could be induced by Peli1-mediated ubiquitination through sequence alignment of amino acids. As shown in revised Figure 7, Peli1-mediated NBS1 ubiquitination was clearly evident in NBS1 WT and K665/683R mutant, but not in NBS1 K686/690R mutant. Results of our microirradiation experiment additionally revealed that NBS1 K686/690R mutant was unable to be retained at DSB sites whereas NBS1 WT and K665/683R mutant were efficiently retained (revised Figures 7d and 7e). Taken together, these results support the notion that Peli1 is required for DSB-induced ATM activation via NBS1 ubiquitination. Relevant statements are also included on pages 15-16 of the revised manuscript.

Reviewers' comments:

Reviewer #1 (Remarks to the Author):

The authors have performed a number of experiments that satisfactorily address the questions raised in my initial review. One thing I am still puzzling over, however, is the observation in Fig. 1h showing GFP-H2AX relocalization after DNA damage. Previous work has not shown any difference in the rate of exchange of H2AX in irradiated vs non-irradiated cells, so I don't know why the authors would observe what appears to be recruitment of the histone to damage sites. Generally it is thought that H2AX is phosphorylated in its preexisting location in chromatin, not that the histone variant is recruited to damage sites. Looking at this issue in the literature, there do seem to be some discrepancies about this and general concern that lasers of different wavelength and power inflict different types of damage, somehow affecting histone exchange rates (see summary of this in PMID: 22704343). At least some discussion of this should be added to manuscript because, while the data appears to be clear, I don't think the explanation of what is happening is very clear to a casual reader.

Reviewer #2 (Remarks to the Author):

With their revisions, the study by Ha and colleagues reporting a role for the E3 ubiquitin ligase Peli1 in the response to DNA double-strand breaks (DSBs) has been significantly improved. However, as described below, a number of conclusions in the manuscript are still not convincingly supported by the data. These issues need to be addressed before publication in Nature Communications can be recommended.

Specific points:

1. The suppression of Peli1 expression under non-pathological conditions and the lack of pronounced phenotypes of Peli1 deltaE4 mice seem at odds with a key upstream role of this E3 ligase in promoting the response to DSBs. Possible scenarios reconciling these conflicting observations should be discussed.
2. The immunoprecipitation results in Fig. 4h,i do not demonstrate direct interaction of Peli1 with the phosphorylated form of H2AX. The text should be corrected accordingly. That the FHA domain of Peli1 directly recognizes pS139 in H2AX is questionable in any case, given the established specificity of FHA domains for phospho-threonine. It is impossible to derive meaningful conclusions from the recruitment data shown in Fig. 4g, given the aberrant behaviour of ectopically expressed GFP-H2AX (accumulation in nucleoli, recruitment to and/or dissociation from laser micro-irradiation-generated DSBs, depending on the status of S139).
3. I have considerable reservations about the authors' conclusion that Peli1 mainly modifies NBS1 with K63-linked ubiquitin chains. The K63 ubiquitination data in Fig. 6d are far from convincing, and the new data in Fig. 6e only show a marginal decrease in NBS1 ubiquitination upon Ubc13 knockdown, suggesting that the bulk of ubiquitin chains on NBS1 are not K63-linked. The new data in Fig. 6f do not help, as the in vitro ubiquitination reactions were apparently performed in the absence of Mms2 or Uev1a, the Ubc13 partner proteins that are essential for its ability to catalyze K63 chain formation. Therefore, it is unclear exactly what Fig. 6f contributes to the manuscript. In the absence of more convincing data, statements on the nature of Peli1-generated ubiquitin conjugates on NBS1 (which seems to be of limited overall importance for the model) should be toned down or removed altogether. Finally, all NBS1 ubiquitination studies were performed with overexpressed Peli1; to better evaluate the relative importance of Peli1, the authors should monitor the impact of depleting endogenous Peli1 on DNA damage-induced NBS1 ubiquitination.

4. It is not clear whether the NBS1 K686/690R mutant is deficient for recruitment to DSBs due to the lack of Peli1-mediated ubiquitination or because these point mutations disrupt the functionality of the ATM interacting motif per se. To further clarify this, complementation experiments with wild-type or inactive forms of ectopically expressed Peli1 could be performed to test whether its E3 ligase activity is required for rescuing NBS1 recruitment to DSBs in shPeli1-transfected cells (Fig. 8e).

Responses to reviewers

Reviewer #1

The authors have performed a number of experiments that satisfactorily address the questions raised in my initial review. One thing I am still puzzling over, however, is the observation in Fig. 1h showing GFP-H2AX relocalization after DNA damage. Previous work has not shown any difference in the rate of exchange of H2AX in irradiated vs non-irradiated cells, so I don't know why the authors would observe what appears to be recruitment of the histone to damage sites. Generally it is thought that H2AX is phosphorylated in its preexisting location in chromatin, not that the histone variant is recruited to damage sites. Looking at this issue in the literature, there do seem to be some discrepancies about this and general concern that lasers of different wavelength and power inflict different types of damage, somehow affecting histone exchange rates (see summary of this in PMID: 22704343). At least some discussion of this should be added to manuscript because, while the data appears to be clear, I don't think the explanation of what is happening is very clear to a casual reader.

Even if phosphorylated H2AX (γ H2AX) was detected at microirradiated sites (Fig. 1e), an initially bleached-out signal of GFP in the context of GFP-fused H2AX (H2AX-GFP) by microirradiation was accumulated at damaged sites (Fig. 1h). We agree with this reviewer that it is widely accepted that H2AX is phosphorylated in its pre-existing location in chromatin, and that the histone variant is not recruited to damaged sites. Our additional experiments on the ectopic expression of H2AX-GFP in H2AX-depleted cells revealed that the signal of γ H2AX was accumulated at DSB sites by microirradiation (Supplementary Fig. 4). This discrepancy of ectopic H2AX-GFP accumulation likely relates to the variability of recruitment to and dissociation from laser microirradiation-generated damage sites, including double-strand breaks, single-strand breaks, and base damages. We have provided additional data in the re-revised Supplementary Fig. 4 and relevant description on page 8.

Reviewer #2

With their revisions, the study by Ha and colleagues reporting a role for the E3 ubiquitin ligase Peli1 in the response to DNA double-strand breaks (DSBs) has been significantly improved. However, as described below, a number of conclusions in the manuscript are still not convincingly supported by the data. These issues need to be addressed before publication in Nature Communications can be recommended.

Specific points:

1. The suppression of Peli1 expression under non-pathological conditions and the lack of pronounced phenotypes of Peli1 deltaE4 mice seem at odds with a key upstream role of this E3 ligase in promoting the response to DSBs. Possible scenarios reconciling these conflicting observations should be discussed.

We agree with this reviewer's comment. We have already shown that the loss of Peli1 led to severe chromosomal defects, such as the extensive rates of DSBs formation (Fig 1a), and the marked reduction of survival rates seen in Peli1^{-/-} mice at six months of age (Supplementary Figs. 5c-5d). Relevant statements had been added to the previously revised version of the manuscript.

2. The immunoprecipitation results in Fig. 4h,i do not demonstrate direct interaction of Peli1 with the phosphorylated form of H2AX. The text should be corrected accordingly. That the FHA domain of Peli1 directly recognizes pS139 in H2AX is questionable in any case, given the established specificity of FHA domains for phospho-threonine. It is impossible to derive meaningful conclusions from the recruitment data shown in Fig. 4g, given the aberrant behaviour of ectopically expressed GFP-H2AX (accumulation in nucleoli, recruitment to and/or dissociation from laser micro-irradiation-generated DSBs, depending on the status of S139).

We have taken this reviewer's comment and rephrased the text accordingly (re-revised

manuscript pages 12 and 20).

3. *I have considerable reservations about the authors' conclusion that Peli1 mainly modifies NBS1 with K63-linked ubiquitin chains. The K63 ubiquitination data in Fig. 6d are far from convincing, and the new data in Fig. 6e only show a marginal decrease in NBS1 ubiquitination upon Ubc13 knockdown, suggesting that the bulk of ubiquitin chains on NBS1 are not K63-linked. The new data in Fig. 6f do not help, as the in vitro ubiquitination reactions were apparently performed in the absence of Mms2 or Uev1a, the Ubc13 partner proteins that are essential for its ability to catalyze K63 chain formation. Therefore, it is unclear exactly what Fig. 6f contributes to the manuscript. In the absence of more convincing data, statements on the nature of Peli1-generated ubiquitin conjugates on NBS1 (which seems to be of limited overall importance for the model) should be toned down or removed altogether. Finally, all NBS1 ubiquitination studies were performed with overexpressed Peli1; to better evaluate the relative importance of Peli1, the authors should monitor the impact of depleting endogenous Peli1 on DNA damage-induced NBS1 ubiquitination.*

In response to the reviewer's comments, to further examine the specificity of NBS1 ubiquitination by Peli1, we performed a series of ubiquitination assays using siRNAs targeting Peli1 (re-revised Fig. 6d) and UBC13 (re-revised Fig. 6f), and three Peli1 mutants, H313A (HA), C336A (CA), and S121A/T127A (AA) (re-revised Fig. 6e). Our new data further support Peli1 mediation of K63-linked ubiquitination of NBS1. We have included new data in the re-revised Figs. 6d-6f, and relevant descriptions on pages 14-15.

4. *It is not clear whether the NBS1 K686/690R mutant is deficient for recruitment to DSBs due to the lack of Peli1-mediated ubiquitination or because these point mutations disrupt the functionality of the ATM interacting motif per se. To further clarify this, complementation experiments with wild-type or inactive forms of ectopically expressed Peli1 could be performed to test whether its E3 ligase activity is required for rescuing NBS1 recruitment to DSBs in shPeli1-transfected cells (Fig. 8e).*

As recommended by this reviewer, we performed additional experiments by transfecting

Peli1 WT and an HA mutant, in combination with GFP-fused NBS1 into endogenous Peli1-depleted cells (Peli1 3'UTR siRNA-transfected cells) and monitored the NBS1 recruitment to DSB sites. As shown in the re-revised Fig. 8e, the recruitment of NBS1 to DSB sites was clearly evident by the expression of ectopic Peli1 WT, but not Peli1 HA mutant in endogenous Peli1-depleted cells, supporting the requirement of Peli1 for DSB-induced ATM activation via NBS1 ubiquitination (re-revised manuscript page 17).

Reviewers' comments:

Reviewer #1 (Remarks to the Author):

It appears that the revised manuscript is contradictory to itself with respect to GFP-H2AX accumulation. Now the authors show the absence of GFP-H2AX staining at 10 minutes post irradiation (the negative line shown in Supp. Fig. 4, although even the data in this figure is not consistent), while the main figure shows positive staining from 2 to 5 min. This does not resolve the problem; in fact now it is more difficult to understand. Is the difference here due to the time? If so then the authors should show a complete time course in the main figure from 0 through 10 min and longer. In addition, the main text needs to actually explain this and discuss the discrepancy between the data shown and other studies in this field. If this is due to differences in laser power and lower power was used to generate the data in the Supp. figure then there would be serious concerns about the relevance of all the laser-related data shown in the main figures.

Reviewer #2 (Remarks to the Author):

The authors have satisfactorily addressed my remaining concerns, and I therefore support publication of the revised manuscript in Nature Communications.

Response to reviewers' comments:

Reviewer #1

It appears that the revised manuscript is contradictory to itself with respect to GFP-H2AX accumulation. Now the authors show the absence of GFP-H2AX staining at 10 minutes post irradiation (the negative line shown in Supp. Fig. 4, although even the data in this figure is not consistent), while the main figure shows positive staining from 2 to 5 min. This does not resolve the problem; in fact now it is more difficult to understand. Is the difference here due to the time? If so then the authors should show a complete time course in the main figure from 0 through 10 min and longer. In addition, the main text needs to actually explain this and discuss the discrepancy between the data shown and other studies in this field. If this is due to differences in laser power and lower power was used to generate the data in the Supp. figure then there would be serious concerns about the relevance of all the laser-related data shown in the main figures.

We apologize for causing this confusion. We agree with the reviewer's opinion. It is generally accepted that H2AX is phosphorylated by ATM, ATR, and DNA-PK in its pre-existing location in damaged chromatins, not that a histone variant H2AX is recruited or accumulated to damage sites.

We previously showed the unexpected result of H2AX-GFP relocalization (accumulation) at laser microirradiation sites, even though an endogenous H2AX was phosphorylated at DSB sites. We also showed that Peli1 as well as damage signalling and repair factors were recruited at damage sites in the same condition of micro-irradiation and endonuclease FokI experiments.

Again, we carefully examined why an ectopically expressed H2AX-GFP appeared to recruit at damage sites. We constructed N-terminal or C-terminal GFP-fused H2AX (GFP-H2AX or H2AX-GFP) and monitored live cell imaging of recruitment of these constructs at DSB sites after microirradiation in HeLa H2AX KO cells (please see the figure for reviewer only). Initially bleached-out GFP signals of N-terminal and C-terminal fused H2AX mainly localized in nucleus were immediately recovered at laser stripes and maintained up to 10 min, but not accumulated at DSB sites. On the other hand, we observed initial accumulation of H2AX-GFP at DSB sites mainly detected in nucleoli. However, the accumulation of H2AX-GFP gradually diminished in a time-dependent manner. We observed similar phenotype of other N-terminal or C-terminal GFP-fused H2AXs at 10 min after microirradiation whereas the bleached-out GFP-H2AX expressed in nucleoli was recovered but not accumulated at DSB sites.

Next, the pattern of GFP-H2AX or H2AX-GFP at DSB sites at 10 min post-microirradiation was evaluated by anti-H2AX and anti- γ H2AX staining. As expected, H2AX was not evicted from DSB sites and γ H2AX was properly detected at DSB sites after GFP-H2AX and H2AX-GFP were transfected into HeLa H2AX KO cells (please see the figure for reviewer only). We also found that GFP-H2AX and H2AX-GFP constructs localized in nucleoli were not detected by anti-H2AX antibody whereas these constructs localized in nuclear were stained with anti-H2AX antibody. These results indicate that both GFP-fused H2AXs highly expressed in nucleoli were not functional H2AX. We are still puzzling over why H2AX-GFP highly expressed in nucleoli is transiently accumulated at DSB sites and then disappears at 9-10 min after microirradiation even though the signal of γ H2AX is detected similarly in other GFP-fused H2AXs.

Thus, we thought it would be better to remove the data of abnormal accumulation of H2AX-

GFP in Fig. 1h and 1i without affecting the overall conclusion of this study. This will make sure that readers do not become confused. However, if the reviewer and editorial board want the original data, we will include these data again.

In the meantime, we would like to make further revision to the related issues in Fig. 4g. We have shown H2AX phosphorylation dependency of Peli1 recruitment at DSB sites using H2AX-GFP. However, we are aware that this data is insufficient to clarify the clear dependency of Peli1 recruitment at DSB sites in H2AX-phosphorylation dependent manner because of the abnormal phenotype of H2AX-GFP in nucleoli. Thus, we performed an additional experiment using other versions of H2AX constructs (FLAG-H2AX wild type, S139E and S139A mutants) in HeLa H2AX KO cells. As presented in the re-revised Fig. 4g, endogenous Peli1 was properly recruited at DSB sites in HeLa wild type cells whereas the recruitment of Peli1 at DSB sites was disrupted in HeLa H2AX KO cells. The ectopic expression of FLAG-H2AX wild type and S139E mutant clearly rescued the recruitment of Peli1 at DSB sites whereas S139A mutant failed to do so. Again, expression patterns of FLAG-H2AX constructs did not show any aberrant localization in nucleoli. We have placed the new data in Fig. 4g.

Figure for reviewer only (Fig. 1h-related data)

a

b

Figure legend. The patterns of GFP-fused H2AX constructs at DSB sites after microirradiation.

(a) HeLa H2AX KO cells were transfected with N-terminal or C-terminal GFP-fused H2AX (GFP H2AX or H2AX-GFP) expressing plasmids. At 48 hr post-transfection, cells were subjected to laser microirradiation. Laser stripes were examined at indicated time point (m). Scale bar, 10 μ m.

(b) GFP-fused H2AX constructs were transfected as described in a. Cells were then fixed at 10 min after micro-irradiation and immunostained with indicated antibodies. Scale bar, 10 μ m.

REVIEWERS' COMMENTS:

Reviewer #1 (Remarks to the Author):

The issues with the previous versions of Fig. 4 and the supplemental data related to it are now much more clear. I agree that it is better to avoid the GFP fusions of H2AX that show strong nucleolar staining. It is likely that these versions of the histone are impaired in some way. The revisions have satisfied my previous issues.